# Linked dimensions of psychopathology and connectivity in functional brain networks

Cedric Huchuan Xia[1], Zongming Ma[2], Rastko Ciric[1], Shi Gu[1,3,4], Richard F. Betzel[3], Antonia N. Kaczkurkin[1], Monica E. Calkins[1], Philip A. Cook[5], Angel García de la Garza[1], Simon N. Vandekar[6], Zaixu Cui[1], Tyler M. Moore[1], David R. Roalf[1], Kosha Ruparel[1], Daniel H. Wolf[1], Christos Davatzikos[5], Ruben C. Gur[1,5], Raquel E. Gur[1,5], Russell T. Shinohara[6], Danielle S. Bassett [3,7] & Theodore D. Satterthwaite [1]

Neurobiological abnormalities associated with psychiatric disorders do not map well to existing diagnostic categories. High co-morbidity suggests dimensional circuit-level abnormalities that cross diagnoses. Here we seek to identify brain-based dimensions of psychopathology using sparse canonical correlation analysis in a sample of 663 youths. This analysis reveals correlated patterns of functional connectivity and psychiatric symptoms. We find that four dimensions of psychopathology – mood, psychosis, fear, and externalizing behavior – are associated ($r = 0.68$–$0.71$) with distinct patterns of connectivity. Loss of network segregation between the default mode network and executive networks emerges as a common feature across all dimensions. Connectivity linked to mood and psychosis becomes more prominent with development, and sex differences are present for connectivity related to mood and fear. Critically, findings largely replicate in an independent dataset ($n = 336$). These results delineate connectivity-guided dimensions of psychopathology that cross clinical diagnostic categories, which could serve as a foundation for developing network-based biomarkers in psychiatry.

[1] Department of Psychiatry, Perelman School of Medicine, University of Pennsylvania, Philadelphia, PA 19104, USA. [2] Department of Statistics, The Wharton School, University of Pennsylvania, Philadelphia, PA 19104, USA. [3] Department of Bioengineering, School of Engineering and Applied Science, University of Pennsylvania, Philadelphia, PA 19104, USA. [4] Department of Computer Science and Engineering, University of Electronic Science and Technology, Chengdu, Sichuan 611731, China. [5] Department of Radiology, Perelman School of Medicine, University of Pennsylvania, Philadelphia, PA 19104, USA. [6] Department of Biostatistics, Epidemiology, and Informatics, Perelman School of Medicine, University of Pennsylvania, Philadelphia, PA 19104, USA. [7] Department of Electrical and Systems Engineering, School of Engineering and Applied Science, University of Pennsylvania, Philadelphia, PA 19104, USA. Correspondence and requests for materials should be addressed to T.D.S. (email: sattertt@pennmedicine.upenn.edu)

Psychiatry relies on signs and symptoms for clinical decision making, without the use of established biomarkers to aid in diagnosis, prognosis, and treatment selection. It is increasingly recognized that existing clinical diagnostic categories could hinder the search for biomarkers in psychiatry[1], as they are not clearly associated with distinct neurobiological abnormalities[2]. The high co-morbidity among psychiatric disorders exacerbates this problem[3]. Furthermore, studies have demonstrated common structural, functional, and genetic abnormalities across psychiatric syndromes, potentially explaining such co-morbidity[4–6]. This body of evidence underscores the lack of direct mapping between clinical diagnostic categories and the underlying pathophysiology.

This context has motivated the development of the National Institute of Mental Health's Research Domain Criteria, which seek to construct a biologically-grounded framework for psychiatric diseases[7]. In such a model, the symptoms of individual patients are conceptualized as the result of mixed dimensional abnormalities of specific brain circuits. While such a model system is theoretically attractive, it has been challenging to implement in practice due to both the multiplicity of clinical symptoms and the many brain systems implicated in psychiatric disorders.

Network neuroscience is a powerful approach for examining brain systems implicated in psychopathology[8,9]. One network property commonly evaluated is its community structure, or modular architecture. A network module (also called a sub-network or a community) is a group of densely interconnected nodes, which may form the basis for specialized sub-units of information processing. Converging results across data sets, methods, and laboratories provide substantial agreement on large-scale functional brain modules such as the somatomotor, visual, default mode, and fronto-parietal control networks[10–12]. Furthermore, multiple studies documented abnormalities within this modular topology in psychiatric disorders[13,14]. Specifically, evidence suggests that many psychiatric disorders are associated with abnormalities in network modules subserving higher-order cognitive processes, including the default mode and fronto-parietal control networks[15,16].

In addition to such module-specific deficits, studies in mood disorders[17,18], psychosis[14,19], and other disorders[20,21] have reported abnormal interactions between modules that are typically segregated from each other at rest. This is of particular interest as modular segregation of both functional[22,23] and structural[24] brain networks is refined during adolescence, a critical period when many psychiatric disorders emerge. Such findings have led many disorders to be considered "neurodevelopmental connectopathies."[25] Describing the developmental substrates of psychiatric disorders is a necessary step towards early identification of at-risk youth, and might ultimately allow for interventions that "bend the curve" of maturation to achieve improved functional outcomes[26].

Despite the increasing interest in describing how abnormalities of brain network development lead to the emergence of psychiatric disorders, existing studies have been limited in several respects. First, most have adopted a categorical case-control approach, or only examined a single dimension of psychopathology[15], and are therefore unable to capture heterogeneity across diagnoses. Second, dimensional psychopathology derived from factor analyses, including our prior work[27–30], were solely driven by covariance in the clinical symptomatology, rather than being guided by both brain and behavior features. Third, especially in contrast to adult studies, existing work in youth has often used relatively small samples (e.g., dozens of participants). While multivariate techniques allow the examination of both multiple brain systems and

clinical dimensions simultaneously, such techniques usually require large samples[31].

In the current study, we seek to delineate functional network abnormalities associated with a broad array of psychopathology in youth. We have capitalized on a large sample of youth from the Philadelphia Neurodevelopmental Cohort (PNC)[32] by applying a recently-developed machine learning technique called sparse canonical correlation analysis (sCCA)[33]. As a multivariate method, sCCA is capable of discovering complex linear relationships between two high-dimensional datasets[34,35]. It should be noted that the approach of the current study is distinct from prior work discovering biotypes within categories of psychopathology, based purely on imaging features themselves (e.g., functional connectivity[36] and gray matter density[37]). In contrast, we seek to link a broad range of symptoms that are present across categories to individual differences in functional brain networks. Such an approach has been successfully applied in prior work on neurodegenerative diseases[34] as well as normal brain-behavior relationships[35].

Here, we use sCCA to delineate linked dimensions of psychopathology and functional connectivity. As described below, we uncover dimensions of connectivity that are highly correlated with specific, interpretable dimensions of psychopathology. We find that each psychopathological dimension is associated with a distinct pattern of abnormal connectivity, and that all dimensions are characterized by decreased segregation of default mode and executive networks (fronto-parietal and salience). These network features linked to each dimension of psychopathology show expected developmental changes and sex differences. Finally, our results are largely replicated in an independent dataset.

## Results

**Linked dimensions of psychopathology and connectivity.** We sought to delineate multivariate relationships between functional connectivity and psychiatric symptoms in a large sample of youth. To do this, we used sCCA, an unsupervised learning technique that seeks to find correlations between two high-dimensional datasets[33]. In total, we studied 999 participants of ages 8–22 who completed both functional neuroimaging and a comprehensive evaluation of psychiatric symptoms as part of the PNC[27,32] (Table 1 and Fig. 1). Participants in the PNC were recruited from Children's Hospital of Philadelphia pediatric network in the greater Philadelphia area. In this community-based study, participants were not recruited from psychiatric services. As such, the prevalence of screening into specific

### Table 1 Philadelphia neurodevelopmental cohort (PNC)

|  |  | Discovery | Replication | Total |
|---|---|---|---|---|
| n |  | 663 | 336 | 999 |
| Sex | Male | 293 | 155 | 448 |
|  | Female | 370 | 181 | 551 |
| Race | White | 306 | 153 | 459 |
|  | Black | 286 | 141 | 427 |
|  | Other | 71 | 42 | 113 |
| Age | 8–10 | 70 | 40 | 110 |
|  | 11–13 | 125 | 63 | 188 |
|  | 14–16 | 195 | 102 | 297 |
|  | 17–19 | 206 | 100 | 306 |
|  | 20–22 | 58 | 30 | 88 |
|  | >22 | 9 | 1 | 10 |
|  | Mean | 15.82 ± 3.32 | 15.65 ± 3.32 | 15.76 ± 3.32 |

The cross-sectional sample of the PNC has 1601 participants in total. After applying health, structural, and functional imaging quality exclusion criteria (details in Online Methods section), 663 and 336 subjects were included in the final discovery and replication samples, respectively

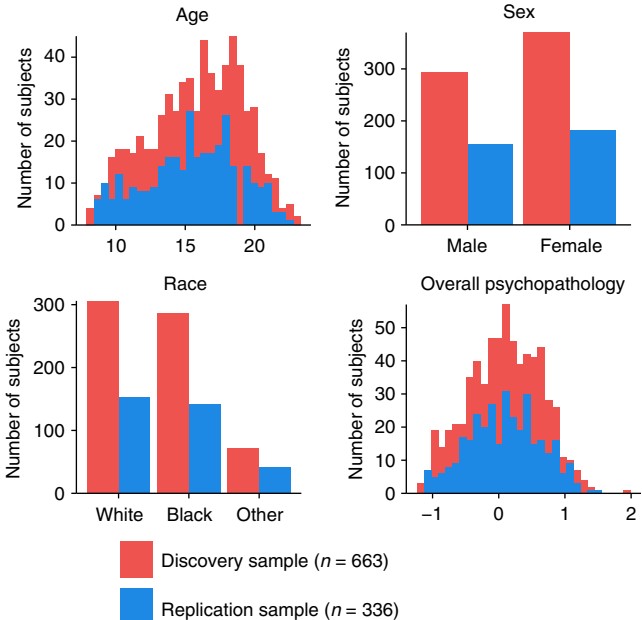

**Fig. 1** Participants demographics. The discovery and replication samples had similar demographic composition, including similar distributions of age, race, sex, and overall psychopathology

psychopathology categories generally aligned with epidemiologically ascertained samples, as previously described[27] (see Supplementary Table 1). We divided this sample into discovery ($n = 663$) and replication datasets ($n = 336$) that were matched on age, sex, race, and overall psychopathology (Fig. 1 and Supplementary Fig. 1). Following pre-processing using a validated pipeline that minimizes the impact of in-scanner motion[38] (see Supplementary Fig. 2, Supplementary Fig. 3 and Supplementary Fig. 4), we constructed subject-level functional networks using a 264-node parcellation system that includes an a priori assignment of nodes to network communities[10] (Fig. 2a–c, e.g., modules or sub-networks; see Online Methods section). Prior to analysis with sCCA, we regressed age, sex, race, and motion out of both the connectivity and clinical data to ensure that these potential confounders did not drive results. As features that do not vary across subjects cannot be predictive of individual differences, we limited our analysis of connectivity data to the top ten percent most variable connections, as measured by median absolute deviation, which is more robust against outliers than standard deviation (Supplementary Fig. 5). The input data thus consisted of 3410 unique functional connections (Fig. 2b) and 111 clinical items (Fig. 2c). The clinical items were drawn from the structured GOASSESS interview[27], and covers a diverse range of psychopathological domains, including mood and anxiety disorders, psychosis-spectrum symptoms, attention-deficit/hyperactivity disorder (ADHD), and other disorders (see details in Supplementary Data 1). Using elastic net regularization ($L1 + L2$) and parameter tuning over both the clinical and connectivity features, sCCA was able to obtain a sparse and interpretable model while minimizing over-fitting (Fig. 2d and Supplementary Fig. 6). Ultimately, sCCA identified specific patterns ("canonical variates") of functional connectivity that were linked to distinct combinations of psychiatric symptoms.

Based on the scree plot of covariance explained (Fig. 3a), we selected the first seven canonical variates for further analysis. Significance of each of these linked dimensions of symptoms and connectivity was assessed using a permutation test, which compares the canonical correlate of each variate to a null distribution built by randomly re-assigning subjects' brain and clinical features (see Online Methods section and Supplementary Fig. 7); False Discovery Rate (FDR) was used to control for type I error rate due to multiple testing. Of these seven canonical variates, three were significant (Pearson correlation $r = 0.71$, $P_{FDR} < 0.001$; $r = 0.70$, $P_{FDR} < 0.001$, $r = 0.68$, $P_{FDR} < 0.01$, respectively) (Fig. 3b), with the fourth showing a trend toward significance ($r = 0.68$, $P_{FDR} = 0.07$, $P_{uncorrected} = 0.04$). Notably, these results were robust to many different methodological choices, including the number of features entered into the initial analysis (Supplementary Fig. 8a), the parcellation system (Supplementary Fig. 8b), and the use of regularization with elastic net versus data reduction with principal component analysis (Supplementary Fig. 8c).

Each canonical variate represented a distinct pattern that relates a weighted set of psychiatric symptoms to a weighted set of functional connections. Inspection of the most heavily weighted clinical symptom for each dimension provided an initial indication regarding their content (Fig. 3c–f). For example, "feeling sad" was the most heavily weighted clinical feature in the first dimension, while "auditory perceptions" was the most prominent symptom in the second. Next, we conducted detailed analyses to describe the clinical and connectivity features driving the observed multivariate relationships.

**Brain-guided dimensions of psychopathology cross clinical diagnostic categories**. To understand the characteristics of each linked dimension, we used a resampling procedure to identify both clinical and connectivity features that were consistently significant across subsets of the data (see Online Methods section and Supplementary Fig. 9). This procedure revealed that 37 out of 111 psychiatric symptoms reliably contributed to at least one of the four dimensions (Fig. 4). Next, we mapped these data-driven items to typical clinical diagnostic categories. This revealed that the features selected by multivariate analyses generally accord with clinical phenomenology. Specifically, despite being selected on the basis of their relationship with functional connectivity, the first three canonical variates delineated dimensions that resemble clinically coherent dimensions of mood, psychosis, and fear (e.g., phobias). The fourth dimension, which was present at an uncorrected threshold, mapped to externalizing behaviors (ADHD and oppositional defiant disorder (ODD)).

While each canonical variate mapped onto coherent clinical features, each dimension contained symptoms from several different clinical diagnostic categories. For example, the mood dimension was comprised of symptoms from categorical domains of depression ("feeling sad" received the highest loading), mania ("irritability"), and obsessive-compulsive disorder (OCD; "recurrent thoughts of harming self or others") (Fig. 4a). Similarly, while the second dimension mostly consisted of psychosis-spectrum symptoms (such as "auditory verbal hallucinations"), two manic symptoms (i.e., "overly energetic" and "pressured speech") were included as well (Fig. 4b). The third dimension was composed of fear symptoms, including both agoraphobia and social phobia (Fig. 4c). The fourth dimension was driven primarily by symptoms of both ADHD and ODD, but also included the irritability item from the depression domain (Fig. 4d). The connectivity-guided clinical dimensions were significantly correlated with, but not identical to, previous factor models such as the bifactor models[28] (see Supplementary Fig. 12). These data-driven dimensions of psychopathology align with clinical phenomenology, but in a dimensional fashion that does not adhere to discrete categories.

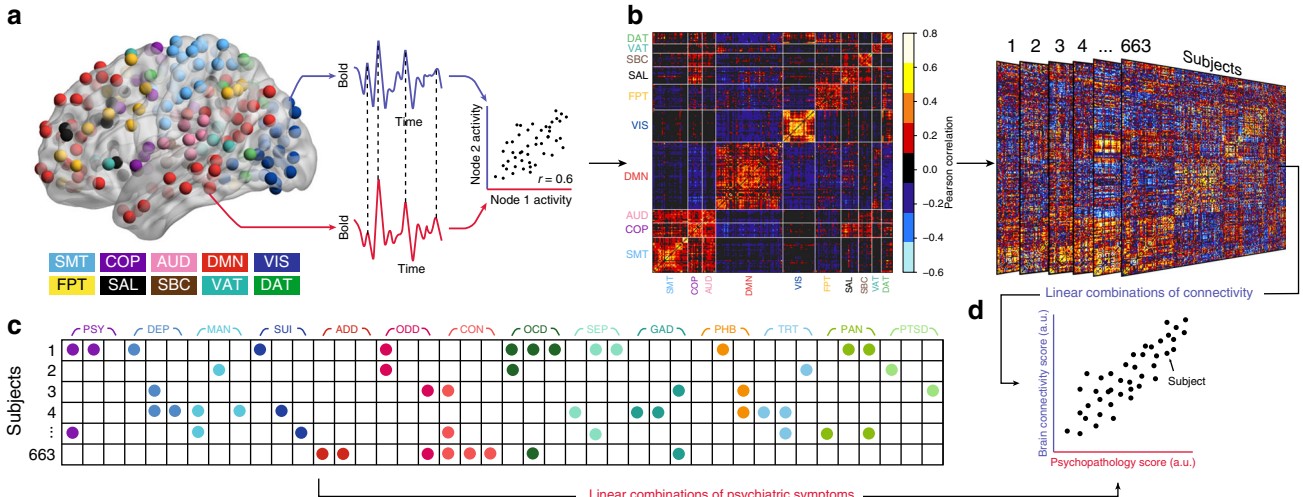

**Fig. 2** Schematic of sparse canonical correlation analysis (sCCA). **a** Resting-state fMRI data analysis schematic and workflow. After preprocessing, blood-oxygen-level dependent (BOLD) signal time series were extracted from 264 spherical regions of interest distributed across the cortex and subcortical structures. Nodes of the same color belong to the same a priori community as defined by Power et al.[10]. **b** A whole-brain, 264 × 264 functional connectivity matrix was constructed for each subject in the discovery sample (n = 663 subjects). **c** Item-level data from a psychiatric screening interview (111 items, based on K-SADS[62]) were entered into sCCA as clinical features (see details in Supplementary Data 1). **d** sCCA seeks linear combinations of connectivity and clinical symptoms that maximize their correlation. A priori community assignment: somatosensory/motor network (SMT), cingulo-opercular network (COP), auditory network (AUD), default mode network (DMN), visual network (VIS), fronto-parietal network (FPT), salience network (SAL), subcortical network (SBC), ventral attention network (VAT), dorsal attention network (DAT), Cerebellar and unsorted nodes not visualized. Psychopathology domains: psychotic and subthreshold symptoms (PSY), depression (DEP), mania (MAN), suicidality (SUI), attention-deficit hyperactivity disorder (ADD), oppositional defiant disorder (ODD), conduct disorder (CON), obsessive-compulsive disorder (OCD), separation anxiety (SEP), generalized anxiety disorder (GAD), specific phobias (PHB), mental health treatment (TRT), panic disorder (PAN), post-traumatic stress disorder (PTSD)

**Common and dissociable patterns of connectivity**. sCCA identified each dimension of psychopathology through shared associations between clinical data and specific patterns of connectivity. Next, we investigated the loadings of connectivity features that underlie each canonical variate. To aid visualization of the high-dimensional connectivity data, we summarized loading patterns according to network communities established a priori by the parcellation system. Specifically, we examined patterns of both within-network and between-network connectivity (Supplementary Fig. 10; see Online Methods section), as this framework has been useful in prior investigations of both brain development[22,23] and psychopathology[17,19,39,40]. This procedure revealed specific patterns of network-level connectivity that were related to the four dimensions of psychopathology (Fig. 5). For example, the mood dimension was characterized by a marked increase in connectivity between the ventral attention and salience networks (Fig. 5a, e, i), while the psychosis dimension received the highest loadings in connectivity between the default mode and executive systems (salience and fronto-parietal networks (Fig. 5b, f, j)). In contrast, increased within-network connectivity of the fronto-parietal network was most evident in the fear dimension (Fig. 5c, g, k). Alterations of the salience system were particularly prominent for the externalizing behavior dimension, including lower connectivity with the default mode network and greater connectivity with the fronto-parietal control network (Fig. 5d, h, l). Quantitatively, the specific loadings of within- and between-network connectivity in each dimension did not significantly correlate with each other (all P > 0.05), demonstrating that each dimension of psychopathology was characterized by a unique pattern of network connectivity.

The results indicate that while each canonical variate was comprised of unique patterns of connectivity, there were several features that were shared across all dimensions. Such findings agree with accumulating evidence for common circuit-level dysfunction across psychiatric syndromes[4,5]. To quantitatively assess such common features, we compared overlapping results against a null distribution using permutation testing (see Online Methods section). This procedure revealed an ensemble of edges that were consistently implicated across all four dimensions. These connections can be mapped to individual nodes, and revealed that the regions most impacted across all dimensions included the frontal pole, superior frontal gyrus, dorsomedial prefrontal cortex, medial temporal gyrus, and amygdala (Fig. 6a). Similar analysis at the level of sub-networks (Fig. 6b) illustrated that abnormalities of connectivity within the default mode and fronto-parietal networks were present in all four psychopathological dimensions (Fig. 6c). Furthermore, reduced segregation between the default mode and executive networks, such as the fronto-parietal and salience systems, was common to all dimensions. These shared connectivity features complement each dimension-specific pattern, and offer evidence for both common and dissociable patterns of connectivity associated with psychopathology.

**Developmental effects and sex differences**. In the above analyses, we examined multivariate associations between connectivity and psychopathology while controlling for participant age. However, given that abnormal brain development is thought to underlie many psychiatric disorders[25,26], we next examined whether connectivity patterns significantly associated with psychopathology differ as a function of age or sex in this large developmental cohort. We repeated the analysis conducted above using connectivity and clinical features, but in this case did not regress out age and sex; race and motion were regressed as prior. Notably, the dimensions derived were quite similar, with highly correlated feature weights (Supplementary Table 2). As in prior work[24,41], developmental associations were examined using

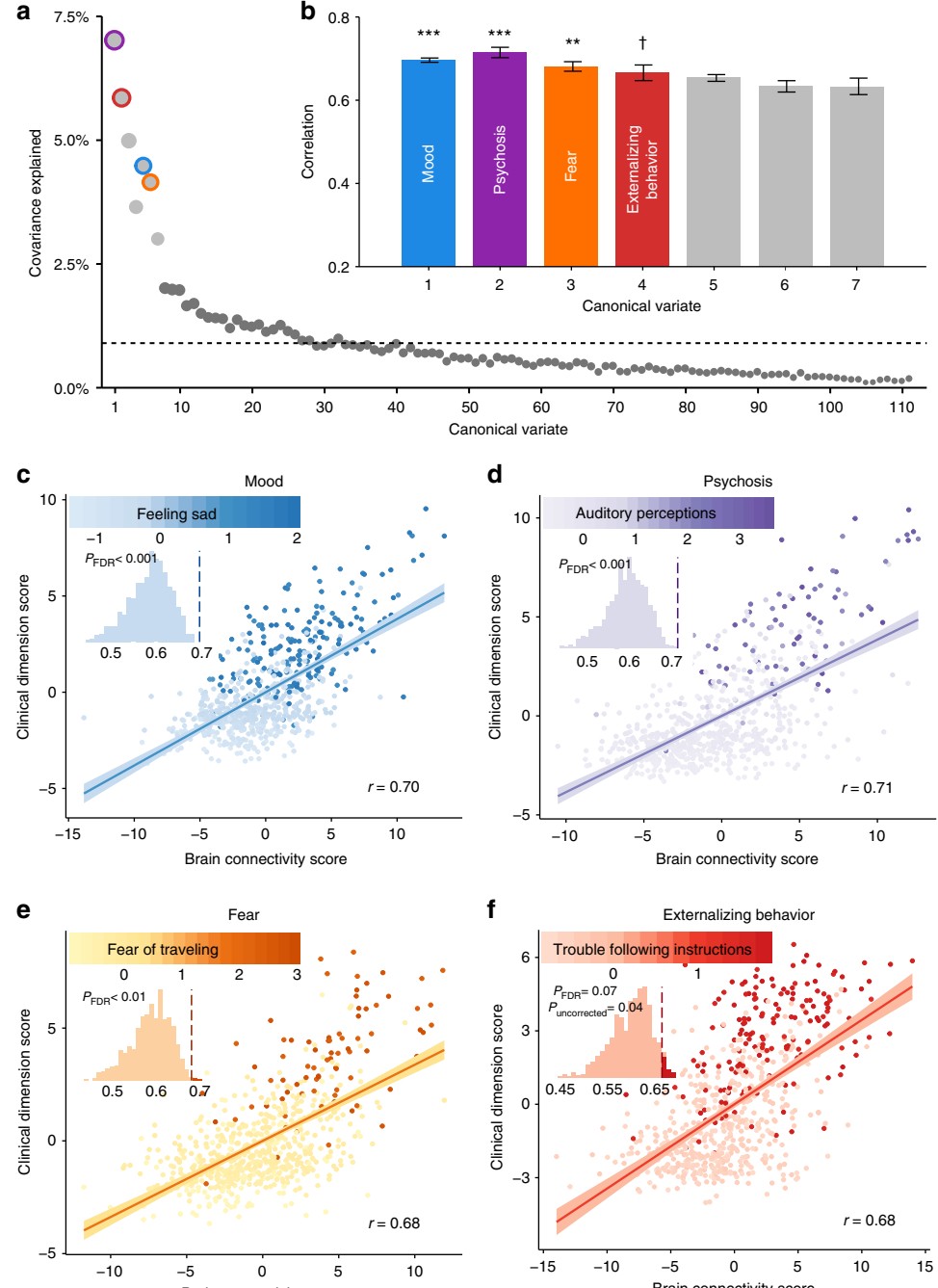

**Fig. 3** sCCA reveals multivariate patterns of linked dimensions of psychopathology and connectivity. **a** The first seven canonical variates were selected based on covariance explained. Dashed line marks the average covariance explained. **b** Three canonical correlations were statistically significant by permutation testing with FDR correction ($q < 0.05$), with the fourth one showing an effect at uncorrected thresholds. Corresponding variates are circled in (**a**). Error bars denote standard error. Dimensions are ordered by their permutation-based $P$ value. **c**–**f** Scatter plots of brain and clinical scores (linear combinations of functional connectivity and psychiatric symptoms, respectively) demonstrate the correlated multivariate patterns of connectomic and clinical features. Colored dots in each panel indicate the severity of a representative clinical symptom that contributed the most to this canonical variate. Each insert displays the null distribution of sCCA correlation by permutation testing. Dashed line marks the actual correlation. ***$P_{FDR} < 0.001$, **$P_{FDR} < 0.01$, †$P_{uncorrected} = 0.04$

generalized additive models with penalized splines, which allows for statistically rigorous modeling of both linear and non-linear effects while minimizing over-fitting. Using this approach, we found that the brain connectivity patterns associated with both mood and psychosis became significantly more prominent with age (Fig. 7a, b, $P_{FDR} < 10^{-13}$, $P_{FDR} < 10^{-6}$, respectively). Additionally, brain connectivity patterns linked to mood and fear were both stronger in female participants than males (Fig. 7c, d, $P_{FDR}$

$< 10^{-8}$, $P_{FDR} < 10^{-7}$, respectively). We did not observe age by sex interaction effects in any dimension.

**Linked dimensions are replicated in an independent sample.** Throughout our analysis of the discovery sample, we used procedures both to guard against over-fitting and to enhance the generalizability of results (regularization, permutation testing,

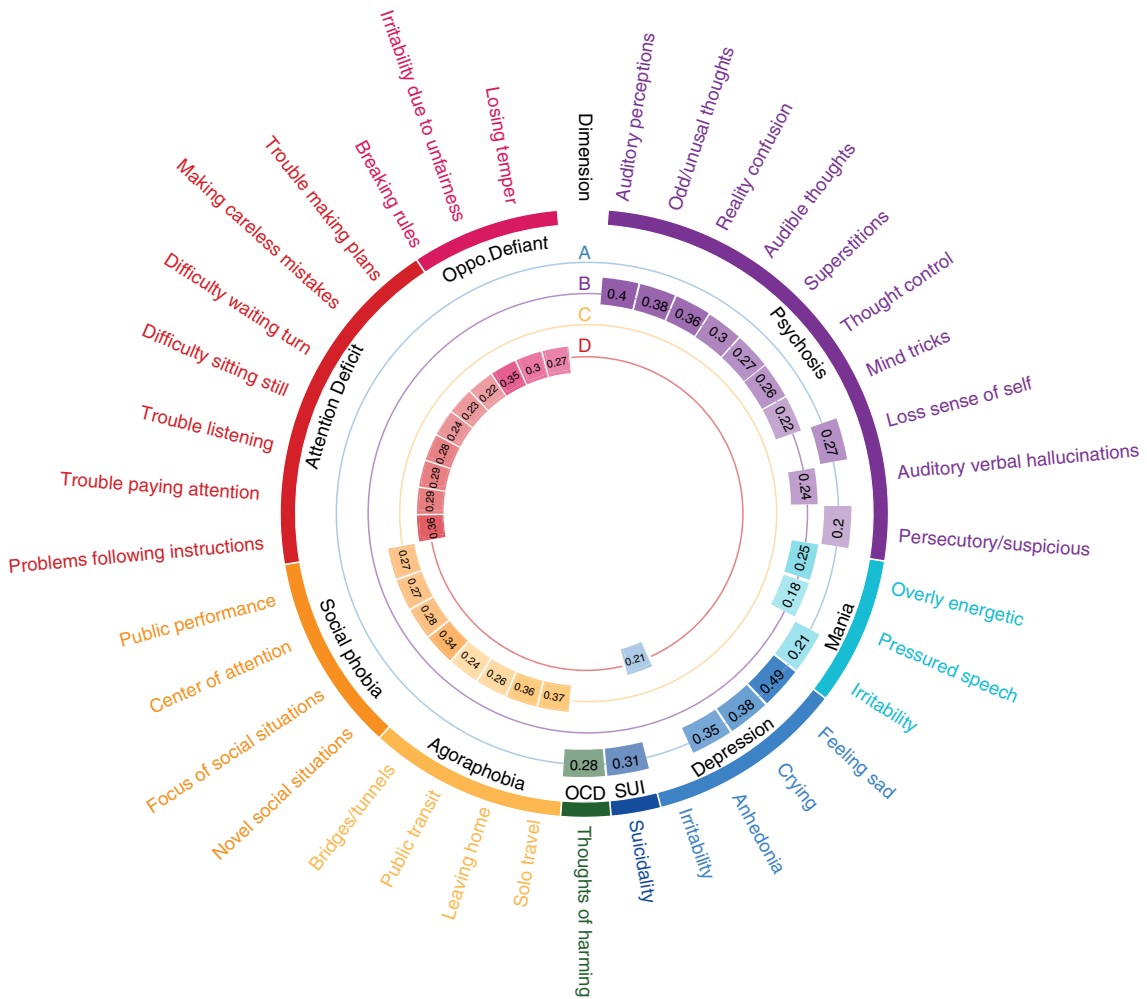

**Fig. 4** Connectivity-informed dimensions of psychopathology cross clinical diagnostic categories. **a** The mood dimension was composed of a mixture of depressive symptoms, suicidality, irritability, and recurrent thoughts of self-harm. **b** The psychotic dimension was composed of psychosis-spectrum symptoms, as well as two manic symptoms. **c** The fear dimension was comprised of social phobia and agoraphobia symptoms. **d** The externalizing behavior dimension showed a mixture of symptoms from attention-deficit and oppositional defiant disorders, as well as irritability from the depression section. The outermost labels are the item-level psychiatric symptoms (see details in Supplementary Data 1). The color arcs represent categories from clinical screening interview and the Diagnostic and Statistical Manual of Mental Disorders (DSM). Numbers in the inner rings represent sCCA loadings for each symptom in their respective dimension. Only loadings determined to be statistically significant by a resampling procedure are shown here

resampling). As a final step, we tested the replicability of our findings using an independent sample, which was left-out from all analyses described above ($n = 336$, Table 1, Fig. 1, and Supplementary Fig. 1). Although this replication sample was half the size of our original discovery sample, sCCA identified four canonical variates that highly resemble the original four linked dimensions of psychopathology. Specifically, the correlations between the clinical loadings in the discovery sample and those in the replication sample were $r = 0.85$ for psychosis ($P_{FDR} < 4.4 \times 10^{-16}$), $r = 0.73$ for externalizing ($P_{FDR} < 4.4 \times 10^{-16}$), $r = 0.59$ for fear ($P_{FDR} = 8.43 \times 10^{-12}$), and $r = 0.23$ for mood ($P_{FDR} = 0.01$). In the replication sample, three out of four dimensions were significant after FDR correction of permutation tests (Fig. 8 and Supplementary Fig. 11). While the bootstrap analysis identified 37 out of 111 symptoms in the discovery sample to consistently contribute to the four linked-dimensions (Fig. 4), the same analysis in the replication sample yielded similar sets of symptoms (80%, 64%, 63%, and 50% overlapping for psychosis, externalizing behavior, fear, and mood, respectively). Additionally, connectivity patterns associated with mood symptoms increased significantly with age ($P_{FDR} = 0.0082$), while connectivity patterns associated

with psychosis symptoms showed a trend towards increasing with age ($P_{uncorrected} = 0.027$, $P_{FDR} = 0.053$). As in the discovery sample, connectivity patterns associated with fear ($P_{FDR} = 0.039$) and mood ($P_{FDR} = 0.0083$) were both elevated in females in the replication sample.

## Discussion

Leveraging a large neuroimaging data set of youth and recent advances in machine learning, we discovered several multivariate patterns of functional connectivity linked to interpretable dimensions of psychopathology that cross traditional diagnostic categories. These patterns of abnormal connectivity were largely replicable in an independent dataset. While each dimension displayed a specific pattern of connectivity abnormalities, loss of network segregation between the default mode and executive networks was common to all dimensions. Furthermore, patterns of connectivity displayed unique developmental effects and sex differences. Together, these results suggest that complex psychiatric symptoms are associated with specific patterns of abnormal connectivity during brain development.

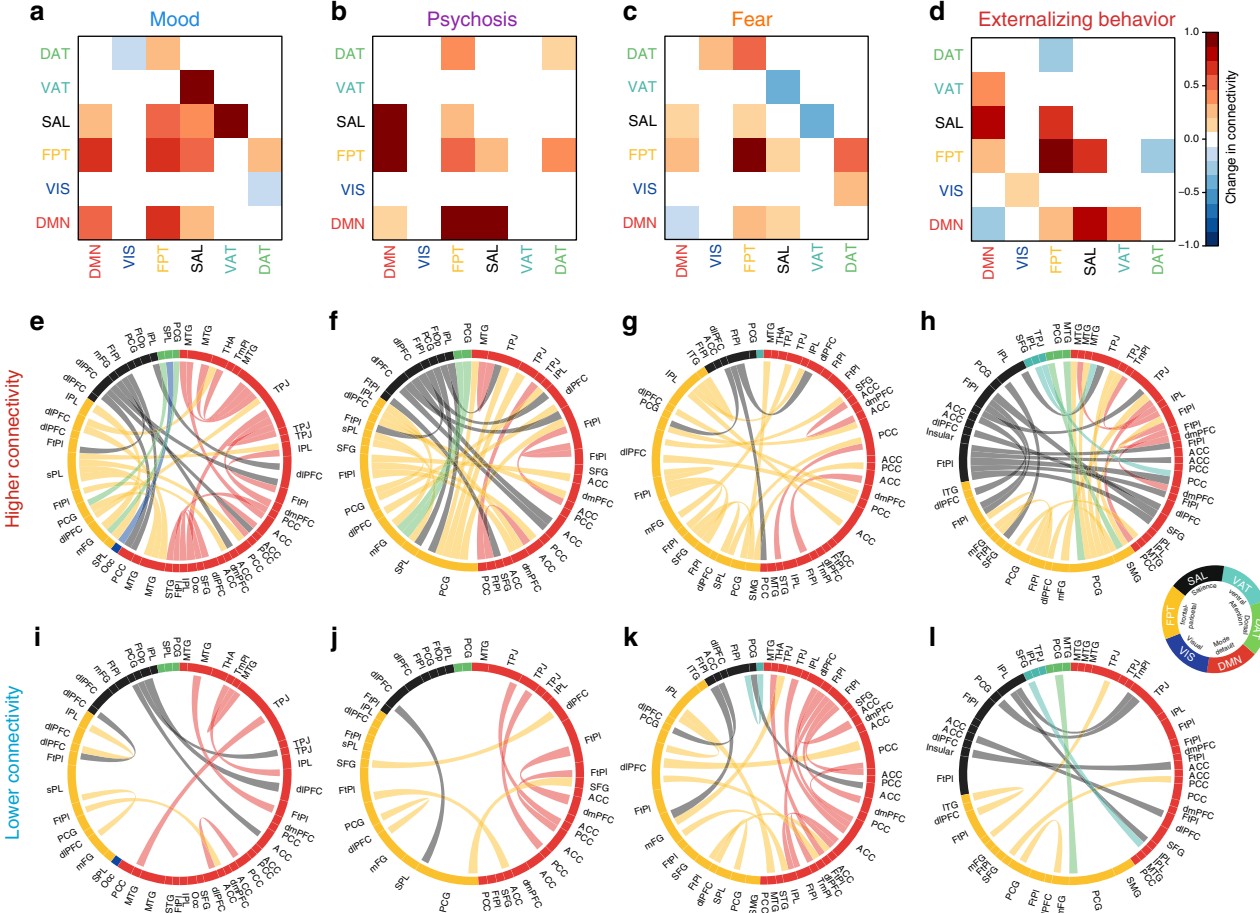

**Fig. 5** Patterns of within- and between-network connectivity contribute to linked psychopathological dimensions. **a–d** Modular (community) level connectivity pattern associated with each psychopathology dimension. Both increased (**e–h**) and diminished (**i–l**) connectivity in specific edges contributed to each dimension of psychopathology. The outer labels represent the anatomical names of nodes. The inner arcs indicate the community membership of nodes. The thickness of the chords represents the loadings of connectivity features

Both the co-morbidity among psychiatric diagnoses and the notable heterogeneity within each diagnostic category suggest that our current symptom-based diagnostic criteria do not "carve nature at its joints"[2]. Establishing biologically-targeted interventions in psychiatry is predicated upon delineation of the underlying neurobiology. This challenge has motivated the NIMH Research Domain Criteria (RDoC) effort, which seeks to link circuit-level abnormalities in specific brain systems to symptoms that might be present across clinical diagnoses[42]. Accordingly, there has been a proliferation of studies that focus on linking specific brain circuit(s) to a specific symptom dimension or behavioral measure across diagnostic categories[43,44]. However, by focusing on a single behavioral measure or symptom domain, many studies ignore the co-morbidity among psychiatric symptoms. A common way to attempt to evaluate such co-morbidity is to find latent dimensions of psychopathology using factor analysis or related techniques. For example, factor analyses of clinical psychopathology have suggested the presence of dimensions including internalizing symptoms, externalizing symptoms, and psychosis symptoms[27,28]. While such dimensions are reliable, they are drawn entirely from the covariance structure of self-report or interview-based clinical data, and are not informed by neurobiology.

An alternative and increasingly pursued approach is to parse heterogeneity in psychiatric conditions using multivariate analysis of biomarker data such as neuroimaging. For example, researchers have used functional connectivity[36] and gray matter density[37] to study the heterogeneity within major depressive

disorder and psychotic disorders, respectively. However, most studies have principally considered only one or two clinical diagnostic categories, and typically the analytic approach yields discrete subtypes (or "biotypes"). By definition, such a design is unable to discover continuous dimensions that span multiple categories. Further, there is tension between the dimensional schema suggested by RDoC and categorical biotypes; as suggested by RDoC, it seems more plausible that psychopathology in an individual results from a mixture of abnormalities across several brain systems. Finally, unsupervised learning approaches using only imaging data and not considering clinical data may frequently yield solutions that are difficult to interpret, and do not align with clinical experience.

In contrast, in this study we used a multivariate analysis technique – sCCA – that allowed simultaneous consideration of clinical and functional connectivity data in a large sample with diverse psychopathology. This method allowed us to uncover linked dimensions of psychopathology and connectivity that cross diagnostic categories yet remain clinically interpretable. Compared to supervised classification methods (e.g., case-control, or multi-class), where each subject is categorized into one discrete class, unsupervised sCCA overcomes the inherent limitation of using discrete diagnostic categories (such as those provided by the Diagnostic and Statistical Manual of Mental Disorders) and allows continuous dimensions of psychopathology to be present in an individual to a varying degree. In addition, in contrast to "one-view" multivariate studies (such as factor analysis of clinical

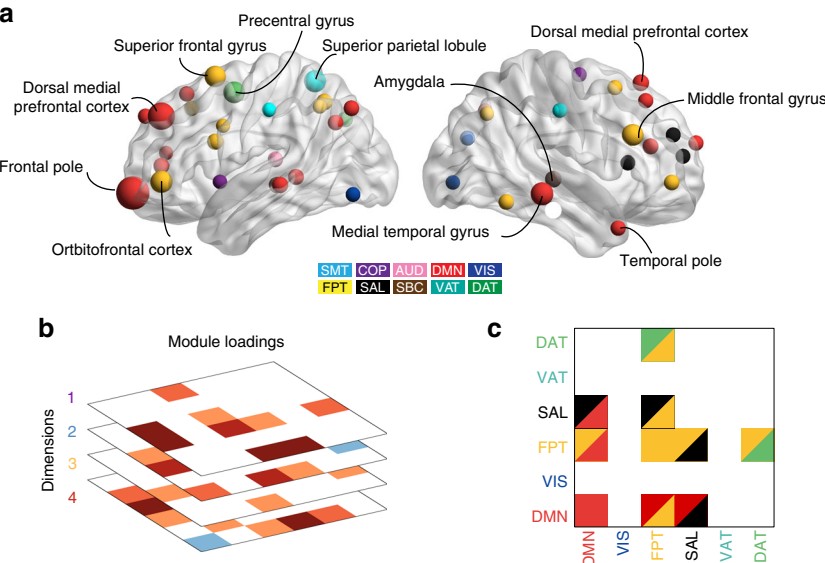

**Fig. 6** Loss of segregation between default mode and executive networks is shared across dimensions. **a** By searching for overlap of edges that contributed significantly to each dimension, we found common edges that were implicated across all dimensions of psychopathology. These were then summarized at a nodal level by the sum of their absolute loadings. Nodes that contributed significantly to every dimension included the frontal pole, superior frontal gyrus, dorsomedial prefrontal cortex, medial temporal gyrus, and amygdala. **b** Results of a similar analysis conducted at the module level. **c** Loss of segregation between the default mode and executive networks was shared across all four dimensions

data or clustering of imaging data)[27,28], the sCCA-derived clinical dimensions were explicitly selected on the basis of co-varying signals that were present as both individual differences of connectivity and clinical symptoms. Such an unsupervised "two-view" approach has been successfully applied in studies of neurodegenerative diseases[34] and normal brain-behavior relationships[35]. In this dimensional, trans-diagnostic approach, the psychopathology of an individual is represented as a mixture of dimensional brain circuit abnormalities, which together produce a specific combination of psychiatric symptoms.

Notably, the brain-driven dimensions described here incorporated symptoms across several diagnostic categories while remaining congruent with prevailing models of psychopathology. For example, the mood dimension was composed of items from five sections of the clinical interview: depression, mania, OCD, suicidality, and psychosis-spectrum. Despite disparate origins, the content of the items forms a clinically coherent picture, including depressed mood, anhedonia, loss of sense of self, recurrent thoughts of self-harm, and irritability. Notably, symptoms of irritability were also significantly represented in the externalizing behavior dimension, suggesting that irritability may have heterogeneous, divergent neurobiological antecedents. The fear dimension, on the other hand, represents a more homogeneous picture of various types of phobias (e.g. social phobia and agoraphobia), that had little overlap with other categorical symptoms. Finally, the psychosis dimension (which was only significant in the discovery sample) was mainly comprised of psychotic symptoms, but also included symptoms of mania. This result accords with studies demonstrating shared inheritance patterns of schizophrenia and bipolar disorder, and findings that specific common genetic variants increase risk of both disorders[45]. Instead of averaging over many clinical features within a diagnostic category, sCCA selected specific items that were most tightly linked to patterns of connectivity. These groups of symptoms remained highly interpretable, and were largely reproducible in the replication data set.

Each of the clinical dimensions identified was highly correlated with patterns of dysconnectivity. These patterns were summarized

according to their location between and within functional network modules, which has been a useful framework for understanding both brain development and psychopathology[19,23]. While each dimension of psychopathology was associated with a unique pattern of dysconnectivity, one of the most striking findings to emerge was evidence that reduction of functional segregation between the default mode and fronto-parietal networks was a common feature of all dimensions. The exact connections implicated in each dimension might vary, but permutation-based analyses demonstrated that loss of segregation between these two networks was present in all four dimensions. Fox et al.[46] originally demonstrated that the default mode network is anti-correlated with task-positive functional brain systems including the fronto-parietal network. Furthermore, studies of brain maturation have shown that age-related segregation of functional brain modules is a robust and reproducible finding regarding adolescent brain development[23,24]. As part of this process, connections within network modules strengthen and connections between two network modules weaken. This process is apparent using functional connectivity[22,23] as well as structural connectivity[24]. Notably, case-control studies of psychiatric disorders in adults have found abnormalities consistent with a failure of developmental network segregation, in particular between executive networks, such as the fronto-parietal and salience networks, and the default mode network[47]. Using a purely data-driven analysis, our results support the possibility that loss of segregation between the default mode and executive networks may be a common neurobiological mechanism underlying vulnerability to a wide range of psychiatric symptoms, lending new evidence for the triple-network model of psychiatric disorders[48,49].

In addition to such common abnormalities that were present across dimensions, each dimension of psychopathology was associated with a unique, highly correlated pattern of dysconnectivity. For example, connectivity features linked to the mood dimension included hyper-connectivity within the default mode, fronto-parietal and salience networks. These dimensional results from a multivariate analysis are remarkably consistent with prior work, which has provided evidence of default mode hyper-

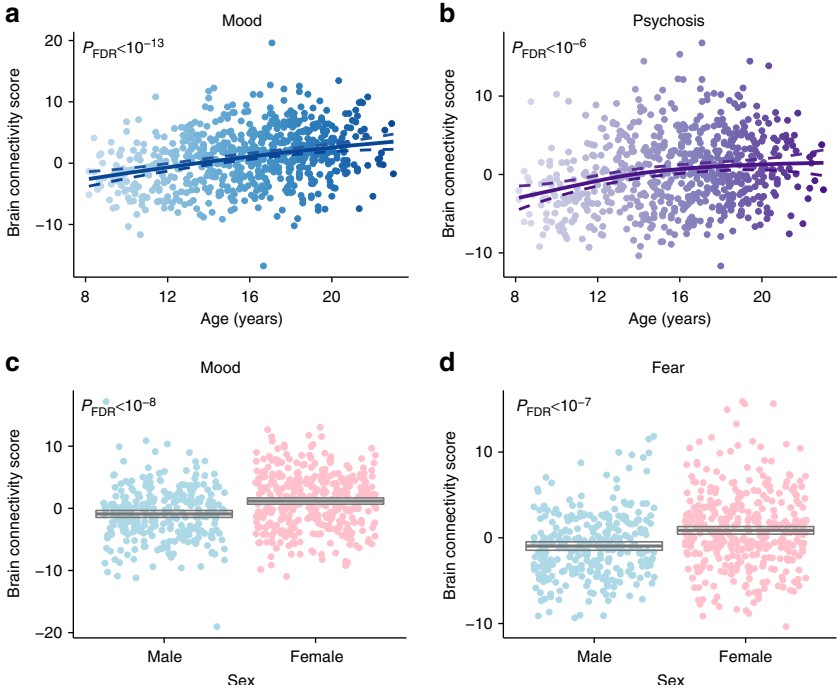

**Fig. 7** Developmental effects and sex differences are concentrated in specific dimensions. Connectivity patterns associated with both the mood (**a**) and psychosis (**b**) dimensions increased significantly with age. Additionally, connectivity patterns associated with both the mood (**c**) and fear (**d**) dimensions were significantly more prominent in females than males. Multiple comparisons were controlled for using the False Discovery Rate ($q < 0.05$). Dashed lines and boxes indicate the 95% confidence interval

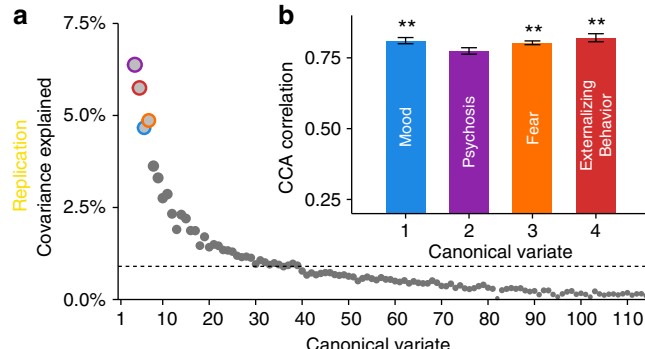

**Fig. 8** Linked dimensions of psychopathology were replicated in an independent sample. All procedures were repeated in an independent replication sample of 336 participants. **a** The first four canonical variates in the replication sample were selected for further analysis based on covariance explained. Dashed line marks the average covariance explained. **b** The mood, fear, and externalizing behavior dimensions were significant by permutation testing. Corresponding variates are circled in (**a**). Error bars denote standard error. **$P_{FDR} < 0.01$

connectivity using conventional case-control designs and uni-variate analysis[50,51]. However, the data-driven approach used here allowed us to discover a combination of novel connectivity features that was more predictive than traditional univariate association analyses. These features included enhanced connectivity between both the dorsal attention and fronto-parietal networks as well as between the ventral attention and salience networks. The fear, externalizing, and psychosis dimensions were defined by a similar mix between novel features and a convergence with prior studies. Specifically, fear was characterized by weakened connectivity within default mode network, enhanced connectivity within fronto-parietal network, and – in contrast to

mood – decreased connectivity between ventral attention and salience networks. In contrast to other dimensions, externalizing behavior exhibited increased connectivity in the visual network and decreased connectivity between fronto-parietal and dorsal attention networks. Finally, the psychosis dimension exhibited stronger connectivity in default mode network and reduced segregation from executive networks (fronto-parietal and salience). Notably, while prior studies have focused on the central role of default mode dysconnectivity in schizophrenia[52] with mixed evidence for hyper-connectivity[53] and hypo-connectivity[54], in the present data the effect within default mode network itself was not nearly as strong as its reduced segregation from the executive networks. Indeed, this finding is consistent with recent data that in psychosis the disruption of segregation between the default mode and task positive networks is a more consistent feature than dysconnectivity within the default mode itself [55].

Importantly, each of these dimensions was initially discovered while controlling for the effects of age and sex. However, given that many psychiatric symptoms during adolescence show a clear evolution with development[56] and marked disparities between males and females[57], we evaluated how the connectivity features associated with each dimension were correlated with age and sex. We found that the patterns of dysconnectivity that linked to mood and psychosis symptoms strengthened with age during the adolescent period. This finding is consistent with the well-described clinical trajectory of both mood and psychosis disorders, which often emerge in adolescence and escalate in severity during the transition to adulthood[58]. In contrast, no age effects were found for externalizing or fear symptoms, which are typically present earlier in childhood and have a more stable time-course[59]. Additionally, we observed marked sex differences in the patterns of connectivity that linked to mood and fear symptoms, with these patterns being more prominent in females across the age range studied. This result accords with data from large-scale epidemiological studies, which have documented a far higher risk

of mood and anxiety disorders in females[60,61]. Despite marked differences in risk by sex (i.e., double in some samples), the mechanism of such vulnerability has been only sparsely studied in the past[32]. The present results suggest that sex differences in functional connectivity may in part mediate the risk of mood and fear symptoms.

Although this study benefited from a large sample, advanced multivariate methods, and replication of results in an independent sample, several limitations should be noted. First, it should be emphasized that our approach did not seek to define biotypes within clinical diagnostic categories in a fully data-driven manner, as in influential prior work[36,37]. Rather, here we sought to provide complementary understanding of heterogeneity by linking symptoms that are present across clinical diagnostic categories to alterations of functional connectivity, uncovering dimensions of psychopathology that are guided by and linked to underlying network abnormalities. However, this approach necessarily is limited by the clinical data being used, in this case item-level data from a structured clinical interview. Although the item-level data used do not explicitly consider clinical diagnostic categories, the items themselves were nonetheless drawn from a standard clinical interview. Incorporating additional data types such as genomics may capture different sources of important biological heterogeneity. Second, while we successfully replicated our findings (except for the psychosis dimension) in an independent sample, the generalizability of the study should be further evaluated in datasets that are acquired in different settings. Third, all data considered in this study were cross-sectional, which has inherent limitations for studies of development. Ongoing follow-up of this cohort will yield informative data that will allow us to evaluate the suitability of these brain-derived dimensions of psychopathology for charting developmental trajectories and prediction of clinical outcome. Fourth, our replication sample was constructed from the PNC data. Using an independently acquired dataset to validate our findings would provide evidence of greater generalizability than splitting the original data into two samples. However, this approach was dictated by the lack of correspondence with clinical instruments used in other large-scale developmental imaging studies. This limitation underscores the need for harmonization of not just imaging data but also clinical measures across studies moving forward. Finally, our current analysis only considered functional connectivity and clinical psychopathology. Future research could incorporate rich multi-modal imaging data, cognitive measures, and genomics.

In summary, in this study we discovered and replicated multivariate patterns of connectivity that are highly correlated with dimensions of psychopathology in a large sample of youth. These dimensions cross traditional clinical diagnostic categories, yet align with clinical experience. Each dimension was composed of unique features of connectivity, while a lack of functional segregation between the default mode network and executive networks was common to all dimensions. Paralleling the clinical trajectory of each disorder and known disparities in prevalence between males and females, we observed both marked developmental effects and sex differences in these patterns of connectivity. As suggested by the NIMH Research Domain Criteria, our findings demonstrate how specific circuit-level abnormalities in the brain's functional network architecture may give rise to a diverse panoply of psychiatric symptoms. Such an approach has the potential to clarify the high co-morbidity between psychiatric diagnoses and the great heterogeneity within each diagnostic category. Moving forward, the ability of these dimensions to predict disease trajectory and response to treatment should be evaluated, as such a neurobiologically-grounded framework could accelerate the rise of personalized medicine in psychiatry.

## Online methods

**Participants**. Resting-state functional magnetic resonance imaging (rs-fMRI) datasets were acquired as part of the Philadelphia Neurodevelopmental Cohort (PNC), a large community-based study of brain development[32]. In total, 1601 participants completed the cross-sectional neuroimaging protocol (Table 1, Fig. 1). One subject had missing clinical data. To create two independent samples for discovery and replication analyses, we performed a random split of the remaining 1600 participants using the CARET package in R. Specifically, using the function create-DataPartition, a discovery sample ($n = 1069$) and a replication sample ($n = 531$) were created that were stratified by overall psychopathology (Supplementary Fig. 1). The two samples were confirmed to also have similar distributions in regards to age, sex, and race (Fig. 1), as well as motion (Supplementary Fig. 2). Overall psychopathology is the general factor score reported previously from factor analysis of the clinical data alone[27,28].

Of the discovery sample ($n = 1069$), 111 were excluded due to gross radiological abnormalities or a history of medical problems that might affect brain function. Of the remaining 958 participants, 45 were excluded for having low quality T1-weighted images, and 250 were excluded for missing rs-fMRI, incomplete image coverage, or excessive motion during the functional scan, which is defined as having an average framewise motion more than 0.20 mm or more than 20 frames exhibiting over 0.25 mm movement (using the Jenkinson calculation[63]). These exclusion criteria produced a final discovery sample consisting of 663 youths (mean age 15.82, SD = 3.32; 293 males and 370 females). Applying the same exclusion criteria to the replication sample produced 336 participants (mean age 15.65, SD = 3.32; 155 males and 181 females). See Table 1 and Fig. 1 for detailed demographics of each sample.

**Psychiatric assessment**. Psychopathology symptoms were evaluated using a structured screening interview (GOASSESS), which has been described in detail elsewhere[27]. To allow rapid training and standardization across a large number of assessors, GOAS-SESS was designed to be highly structured, with screen-level symptom and episode information. The instrument is abbreviated and modified from the epidemiologic version of the NIMH Genetic Epidemiology Research Branch Kiddie-SADS[62]. The psychopathology screen in GOASSESS assessed lifetime occurrence of major domains of psychopathology including psychosis spectrum symptoms, mood (major depressive episode, mania), anxiety (agoraphobia, generalized anxiety, panic, specific phobia, social phobia, separation anxiety), behavioral disorders (oppositional defiant, attention deficit/hyperactivity, conduct), eating disorders (anorexia, bulimia), and suicidal thinking and behavior (Supplementary Table 1). The 111 item-level symptoms used in this study were described in prior factor analysis of the clinical data in PNC[28]. For the specific items, see Supplementary Data 1.

**Image acquisition**. Structural and functional subject data were acquired on a 3T Siemens Tim Trio scanner with a 32-channel head coil (Erlangen, Germany), as previously described[32]. High-resolution structural images were acquired in order to facilitate alignment of individual subject images into a common space. Structural images were acquired using a magnetization-prepared, rapid-acquisition gradient-echo (MPRAGE) T1-weighted sequence ($T_R = 1810$ ms; $T_E = 3.51$ ms; FoV = $180 \times 240$ mm; resolution $0.9375 \times 0.9375 \times 1$ mm). Approximately 6 minutes of task-free functional data were acquired for each subject using a blood oxygen level-dependent (BOLD-weighted) sequence ($T_R = 3000$ ms; $T_E = 32$ ms; FoV = $192 \times 192$ mm; resolution 3 mm isotropic; 124 volumes). Prior to scanning, in order to acclimate

subjects to the MRI environment and to help subjects learn to remain still during the actual scanning session, a mock scanning session was conducted using a decommissioned MRI scanner and head coil. Mock scanning was accompanied by acoustic recordings of the noise produced by gradient coils for each scanning pulse sequence. During these sessions, feedback regarding head movement was provided using the MoTrack motion tracking system (Psychology Software Tools, Inc., Sharpsburg, PA). Motion feedback was only given during the mock scanning session. In order to further minimize motion, prior to data acquisition subjects' heads were stabilized in the head coil using one foam pad over each ear and a third over the top of the head. During the resting-state scan, a fixation cross was displayed as images were acquired. Subjects were instructed to stay awake, keep their eyes open, fixate on the displayed crosshair, and remain still.

**Structural preprocessing**. A study-specific template was generated from a sample of 120 PNC subjects balanced across sex, race, and age bins using the buildtemplateparallel procedure in ANTS[64]. Study-specific tissue priors were created using a multi-atlas segmentation procedure[65]. Subject anatomical images were independently rated by three highly trained image analysts. Any image that did not pass manual inspection was removed from the analysis. Each subject's high-resolution structural image was processed using the ANTS Cortical Thickness Pipeline[66]. Following bias field correction[67], each structural image was diffeomorphically registered to the study-specific PNC template using the top-performing SYN deformation provided by ANTS[68]. Study-specific tissue priors were used to guide brain extraction and segmentation of the subject's structural image[69].

**Functional preprocessing**. Task-free functional images were processed using one of the top-performing pipelines for removal of motion-related artifact[38]. Preprocessing steps included (1) correction for distortions induced by magnetic field inhomogeneities using FSL's FUGUE utility, (2) removal of the 4 initial volumes of each acquisition, (3) realignment of all volumes to a selected reference volume using MCFLIRT[63], (4) removal of and interpolation over intensity outliers in each voxel's time series using AFNI's 3DDESPIKE utility,
 (5) demeaning and removal of any linear or quadratic trends, and (6) co-registration of functional data to the high-resolution structural image using boundary-based registration[70]. The artefactual variance in the data was modelled using a total of 36 parameters, including the six framewise estimates of motion, the mean signal extracted from eroded white matter and cerebrospinal fluid compartments, the mean signal extracted from the entire brain, the derivatives of each of these nine parameters, and quadratic terms of each of the nine parameters and their derivatives. Importantly, our findings are robust to the methodological choice of regressing out global signal (Supplementary Fig. 3 and Supplementary Fig. 4). Both the BOLD-weighted time series and the artefactual model time series were temporally filtered using a first-order Butterworth filter with a passband between 0.01 and 0.08 Hz[71].

**Network construction**. We built a functional connectivity network using the residual timeseries (following de-noising) of all parcels of a common parcellation[10]. The parcellation used in the main analysis consists of 264 spherical nodes of 20 mm diameter distributed across the brain.[10] The a priori communities for this set of nodes were originally delineated using the Infomap algorithm[72] and were replicated in an independent sample. This parcellation was particularly suitable for our analysis as it has been previously used for studying developmental changes in connectivity and network modularity[22] and has been used as part of several studies in this dataset in the past[38,73,74]. As part of the supplementary analysis to demonstrate the robustness of the results independent of parcellation choices (Supplementary Fig. 8), we also constructed networks based on an alternative parcellation developed by Gordon et al.[12]. This set of nodes was derived using edge detection and boundary mapping to define areal parcels. The functional connectivity between any pair of brain regions was operationalized as the Pearson correlation coefficient between the mean activation timeseries extracted from those regions. For each parcellation, an $n \times n$ weighted adjacency matrix encoding the connectome was thus obtained, where n represents the total number of nodes (or parcels) in that parcellation. Community boundaries were defined a priori for each parcellation scheme.

To ensure that potential confounders did not drive the canonical correlations, we regressed out relevant covariates out of the input matrices. Specifically, using the glm and residual.glm functions in R, we regressed age, sex, race, and in-scanner motion out of the connectivity data, and regressed age, sex, and race out of the clinical data. Importantly, we found that the canonical variates derived from regressed and non-regressed datasets were comparable, with highly correlated feature weights (Supplementary Table 2).

**Dimensionality reduction**. Each correlation matrix comprised 34,980 unique connectivity features. We reasoned that since sCCA seeks to capture sources of variation common to both datasets, connectivity features that are most predictive of psychiatric symptoms would be those with high variance across participants. Therefore, to reduce dimensionality of the connectivity matrices, we selected the top edges with the highest median absolute deviation (MAD) (Supplementary Fig. 5). MAD is defined as $\text{median}(|\mathbf{X}_i - \text{median}(\mathbf{X})|)$, or the median of the absolute deviations from the vector's median. We chose MAD as a measurement for variance estimation, because it is a robust statistic, being more resilient to outliers in a data set than other measures such as the standard deviation. To illustrate which edges were selected based on MAD, we visualized the network adjacency matrix with all edges, at 95th, 90th, and 75th percentile (Supplementary Fig. 5c).

An alternative approach for dimensionality reduction is principal component analysis (PCA), from which we selected the top 111 components (explaining 37% of variance) as connectivity features entered into sCCA. As detailed in Supplementary Fig. 8, using PCA yielded similar canonical variates as MAD. We ultimately chose feature selection with MAD because it allowed direct use of individual connectivity strength instead of latent variables (e.g. components from PCA) as the input features to sCCA, thus increasing the interpretability of our results.

**Sparse canonical correlation analysis**. sCCA is a multivariate procedure that seeks maximal correlations between linear combinations of variables in both sets, with regularization to achieve sparsity[33]. In essence, given two matrices, $\mathbf{X}_{n \times p}$ and $\mathbf{Y}_{n \times q}$, where $n$ is the number of observations (e.g., participants), $p$ and $q$ are the number of variables (e.g., clinical and connectivity features, respectively), sCCA involves finding $\mathbf{u}$ and $\mathbf{v}$, which are loading vectors, that maximize cor ($\mathbf{Xu}, \mathbf{Yv}$). Mathematically, this

optimization problem can be expressed as

$$\text{maximize } \mathbf{u}^{\mathrm{T}}\mathbf{X}^{\mathrm{T}}\mathbf{Y}\mathbf{v}, \text{ subject to } ||\mathbf{u}||_2^2 \leq 1, \ ||\mathbf{v}||_2^2 \leq 1, \tag{1}$$
$$||\mathbf{u}||_1 \leq c_1, \ ||\mathbf{v}||_1 \leq c_2.$$

Since both $L^1$ ($||\cdot||_1$) and $L^2$-norm ($||\cdot||_2$) are used, this is an elastic net regularization that combines the LASSO and ridge penalties. The penalty parameters for the $L^2$ norm are fixed for both $\mathbf{u}$ and $\mathbf{v}$ at 1, but those of $L^1$ norm, namely $c_1$ and $c_2$, are set by the user and need to be tuned (see below).

We chose a linear kernel over non-linear implementations of sCCA for two reasons. First, while a more complex model may potentially better fit the data, increased model complexity often results in reduced interpretability. Secondly, a non-linear model may require a larger sample size to accurately estimate the increased number of parameters.

**Grid search for regularization parameters.** We tuned the $L^1$ regularization parameters for the connectivity and the clinical features, respectively (see Supplementary Fig. 6). The range of sparsity parameters are constrained to be between 0 and 1 in the PMA package[33], where 0 indicates the smallest number of features (i.e., highest level of sparsity) and 1 indicates the largest number of features (i.e., lowest level of sparsity). We conducted a grid search in increments of 0.1 to determine the combination of parameters that would yield the highest canonical correlation of the first variate across 10 randomly resampled samples, each consisting of two-thirds of the discovery dataset. Note that the parameters were only tuned on the discovery sample and the same regularization parameters were applied in the replication analysis.

**Permutation testing.** To assess the statistical significance of each canonical variate, we used a permutation testing procedure to create a null distribution of correlations (Supplementary Fig. 7). Essentially, we held the connectivity matrix constant, and then shuffled the rows of the clinical matrix so as to break the linkage of participants' brain features and their symptom features. Then we performed sCCA using the same set of regularization parameters to generate a null distribution of correlations after permuting the input data 1000 times ($B$). As permutation could induce arbitrary axis rotation, which changes the order of canonical variates, or axis reflection, which causes a sign change for the weights, we matched the canonical variates resulting from permuted data matrices to the ones derived from the original data matrix by comparing the clinical loadings ($\mathbf{v}$)[75]. The $P_{\mathrm{FDR}}$ value was estimated as the number of null correlations ($r_i$) that exceeded the average sCCA correlations estimated on the original dataset ($\bar{r}$), with false discovery rate correction (FDR, $q < 0.05$) across the top seven selected canonical variates:

$$P_{\mathrm{permutation}} = \frac{\sum_{i=1}^{B} \begin{cases} 1, \text{ if } r_i \geq \bar{r} \\ 0, \text{ if } r_i < \bar{r} \end{cases}}{B}. \tag{2}$$

In other words, we randomly assigned subjects' clinical features to other subjects' connectivity features, therefore breaking up the internal co-varying structures of the original dataset. The canonical variates resulting from these re-aligned datasets with preserved data distribution will then serve as the null distribution against which the real correlations are compared. The logic is that any significant co-varying relationships will have to be greater than the signals in a permuted data structure.

**Resampling procedure.** To further select features that consistently contributed to each canonical variate, we performed a resampling procedure (Supplementary Fig. 9). In each of 1000 samples, we randomly selected two-thirds of the discovery sample and then randomly replaced the remaining one-third from those two-thirds (similar to bootstrapping with replacement). Similar to the permutation procedure, we matched the corresponding canonical variates from resampled matrices to the original one to obtain a set of comparable decompositions[75]. Features whose 95% and 99% confidence intervals (for clinical and connectivity features, respectively) did not cross zero were considered significant, suggesting that they were stable across different sampling cohorts.

**Network module analysis.** To visualize and understand the high dimensional connectivity loading matrix, we summarized it as mean within- and between-module loadings according to the a priori community assignment of the Power parcellation (Supplementary Fig. 10a)[10]. Specifically, within-module connectivity loading is defined as

$$\frac{\sum_{i,j \in m} 2W_{ij}}{|\mathbf{M}| \times (|\mathbf{M}| - 1)}, \tag{3}$$

where $W_{ij}$ is the sCCA loading of the functional connectivity between nodes $i$ and $j$, which both belong to the same community $m$ in $\mathbf{M}$. The cardinality of the community assignment vector, $|\mathbf{M}|$, represents the number of nodes in each community. Between-module connectivity loading is defined as

$$\frac{\sum_{i \in m, j \in n} W_{ij}}{|\mathbf{M}| \times |\mathbf{N}|}, \tag{4}$$

where $W_{ij}$ is the sCCA loading of the functional connectivity between nodes $i$ and $j$, which belong to community $m$ in $|\mathbf{M}|$ and community $n$ in $|\mathbf{N}|$, respectively.

We used a permutation test based on randomly assigning community memberships to each node while controlling for community size to assess the statistical significance of the mean connectivity loadings (Supplementary Fig. 10b). Empirical P-values were calculated similar to Eq. 2 and were FDR-corrected.

**Analysis of common connectivity features across dimensions.** Each connectivity loading matrix was first binarized based on the presence of a significant edge feature after the resampling procedure in a given canonical variate. All four binarized matrices were then added and thresholded at 4 (i.e. common to all four dimensions), generating an overlapping edge matrix. Statistical significance was assessed by comparing this concordant feature matrix to a null model. The null model was constructed by computing the overlapping edges, repeated 1000 times, of four randomly generated loading matrices, each preserving the edge density of the original loading matrix. Any edge that appeared at least once in the null model was eliminated from further analysis. With only the statistically significant common edge features, we calculated the mean absolute loading in each edge feature across four dimensions as well as the nodal loading strength using Brain Connectivity Toolbox[76] and visualized it with BrainNet Viewer[77] both in MATLAB.

**Analysis of age effects and sex differences.** As previously[24,28,78], generalized additive models (GAMs), using the MGCV package in R, were used to characterize age-related effects and sex differences on the specific dysconnectivity pattern associated with each psychopathology dimension. A GAM is similar to a generalized linear model, where predictors can be replaced by smooth functions of themselves, offering efficient and flexible

estimation of non-linear effects. For each linked dimension $i$, a GAM was fit:

$$\text{Connectivity Score}_i \sim \text{Sex} + \text{s(Age)}\cdot \qquad (5)$$

Additionally, we also separately tested whether age by sex interactions were present.

**Data availability**. The data reported in this paper have been deposited in database of Genotypes and Phenotypes (dbGaP): accession no. [phs000607.v3.p2].

**Code availability**. All analysis code is available here: https://github.com/cedricx/sCCA/tree/master/sCCA/code/final

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

## Acknowledgements

Thanks to Chad Jackson for data management and systems support. This study was supported by grants from National Institute of Mental Health: R01MH107703 (T.D.S.), R01MH112847 (R.T.S. & T.D.S.), R21MH106799 (D.S.B. & T.D.S.), R01MH107235 (R.C. G.), R01MH113550 (T.D.S. & D.S.B.), and R01EB022573 (C.D.). The PNC was supported by MH089983 and MH089924. Additional support was provided by P50MH096891 to R.E.G., R01MH101111 to D.H.W., K01MH102609 to D.R.R., K08MH079364 to M.E.C., R01NS085211 to R.T.S., and the Dowshen Program for Neuroscience. Additionally, D.S.B. acknowledges support from the John D. and Catherine T. MacArthur Foundation, the Alfred P. Sloan Foundation, the ISI Foundation, the Paul Allen Foundation, the Army Research Laboratory (W911NF-10-2-0022), the Army Research Office (Bassett-W911NF-14-1-0679, Grafton-W911NF-16-1-0474, DCIST-W911NF-17-2-0181), the Office of Naval Research, the National Institute of Mental Health (2-R01-DC-009209-11, R01 – MH112847, R01-MH107235, R21-M MH-106799), the National Institute of Child Health and Human Development (1R01HD086888-01), National Institute of Neurological Disorders and Stroke (R01 NS099348), and the National Science Foundation (BCS-1441502, BCS-1430087, NSF PHY-1554488 and BCS-1631550). The content is solely the responsibility of the authors and does not necessarily represent the official views of any of the funding agencies.

## Author contributions

C.H.X., R.C., S.G., A.N.K., M.E.C., A.G., T.M.M., D.R.R., K.R., D.H.W., R.C.G., R.E.G., D. S.B., and T.D.S. designed the research; C.H.X., Z.M., R.C., D.S.B., and T.D.S. performed research; C.H.X., Z.M., R.C., S.G., R.F.B., P.A.C., S.V., Z.C., C.D., R.T.S., D.S.B., and T.D.S. contributed analytic tools; C.H.X., Z.M., R.C., D.S.B., and T.D.S. analyzed data; and C.H.X., Z.M., R.C., S.G., R.F.B., A.N.K., M.E.C., P.A.C., A.G., S.V., Z.C., T.M.M., D.R. R., K.R., D.H.W., C.D., R.C.G. R.E.G., R.T.S., D.S.B., and T.D.S. wrote the paper.

## Additional information

**Competing interests:** R.T.S. has received legal consulting and advisory board income from Genentech/Roche. All other authors (C.H.X., Z.M., R.C., S.G., R.F.B., A.N.K., M.E. C., P.A.C., A.G., S.V., Z.C., T.M.M., D.R.R., K.R., D.H.W., C.D., R.C.G., R.E.G., D.S.B., and T.D.S.) declare no competing interest.

