## [Peer Review File · Nature Communications]

Reviewers' comments:

Reviewer #1 (Remarks to the Author):

This is a data-driven analysis of a large developmental dataset. The aim is to link symptoms of psychopathology with abnormalities in resting-state functional-connectivity MRI using canonical correlation analysis, a dimensionality reduction technique which maximizes correlations between latent variables. The main observed effect is a discovery of strong relationships between specific functional network abnormalities and 3-4 distinct clusters of symptoms (dimensions of psychopathology).

I was impressed with several aspects of this study which clearly make it stand out from the pack. First, the primary dataset is big (n=663) and quite unique in its combination of large size, a detailed psychopathological assessment and resting-state imaging. Second, the authors employed statistically principled machine learning analysis techniques, rather than the more handwavy tools of network neuroscience. Third, the authors have taken care to ensure that their findings are robust, with permutation tests, resampling, and most importantly replication of the main effects in an independent part of their dataset (n=336); this is excellent practice and should be compulsory for the field more generally.

Nonetheless, I have two main concerns. First, I feel that the study is presented in quite a convoluted way. The main results are essentially about the mapping of symptoms to circuits. This is valuable, but cannot in itself allow us to make progress towards a biologically-grounded understanding of psychiatric disorders, despite what the study claims. Second, the processing of functional MRI data is a field fraught with dangers, including global signal regression, motion correction, and vigilance control. The authors have pursued an aggressive strategy to remove sources of artifact from their data, but it is important to show that this strategy has not led to an undue deformation of the (already) fairly low-dimensional signal and the resulting description of their functional networks. I expand on this and other points below.

1. Conceptual formulation

To appreciate the main conceptual concern, consider how the present abstract might be rewritten from the perspective of a more biologically-grounded field, such as respiratory medicine:

"[a] Physiological abnormalities associated with respiratory disorders do not map well to existing diagnostic categories. High co-morbidity and overlapping symptom domains suggest abnormalities that cut across clinical diagnoses. [b] Here we sought to identify respiratory-tract based dimensions of respiratory pathology. We found that several dimensions of respiratory pathology -- rhinorrhea, cough and wheeze -- were highly associated with distinct patterns of pathophysiology. [c] These results delineate physiologically guided dimensions of respiratory pathology that cut across traditional diagnostic categories, which could serve as a foundation for developing objective biomarkers in respiratory medicine."

Consider the problems inherent in these statements. Statement [a] is presented as a foundational limitation that is holding back progress for objective diagnosis. Yet in respiratory medicine no one seems concerned by the fact that the symptom of a runny nose (rhinorrhea) cuts across clinical diagnostic categories; it does so by design, as an end-stage process of distinct upstream mechanisms (diseases). [b] the present study maps psychiatric symptoms to abnormalities of neural circuits, much in the same way as rhinorrhea could be mapped to an inflamed nasal mucosa. This is a valuable contribution as such mappings are largely absent in psychiatry. But it should not be surprising that these clusters of symptoms cut across diagnostic categories; in fact, it is expected that they should do

so! [c] implies that such mappings could serve to develop objective biomarkers of psychiatric disorders, which is arguably not the case here: understanding an association between inflammation and rhinorrhea cannot delineate diseases which give rise to the inflammation. The same seems to hold for psychiatric symptoms and circuit abnormalities.

Put another way, the present study is not a study of psychiatric disorders (upstream), but a study of psychiatric symptoms (downstream). This becomes apparent when the authors find that "features selected by multivariate analyses generally accord with clinical phenomenology". In contrast, consider a related work on biotypes (Clementz et al., 2015) which takes a more upstream view and identifies biomarkers independently of clinical features, albeit in a more restricted disease setting of psychosis. It seems to me that such an upstream approach is more explicitly trying to achieve what the authors set out to do here, namely to classify psychiatric disorders in an objective and quantitative manner.

This distinction is critical, and is a major distinguishing feature of RDoC (Cuthbert and Insel, 2013). "The distinctions between RDoC and the DSM and ICD systems can be captured by seven major points that include both conceptual and practical differences. First, the approach incorporates a strong translational research perspective. ***Rather than starting with symptom-based definitions of disorders and working toward their pathophysiology, RDoC inverts this process. Basic science - in genetics, other areas of neuroscience and behavioral science - serves as the starting point, and disorders are considered in terms of disruptions of the normal-range operation of these systems, with an emphasis on the mechanisms that serve to result in dysfunctions of varying degrees***."

In this context, my suggestion to the authors is to rewrite major portions of the text (in the abstract, introduction, results and discussion) which concern the motivation and the interpretation of their work. They should more clearly motivate and interpret their contributions, by de-emphasizing aspects concerning classification of psychiatric disorders and emphasizing and motivating the rationale for linking symptom groupings with circuit abnormalities.

2. Methods

The authors employ an aggressive artifact correction pipeline, removing 36 temporal degrees of freedom from each 120-dimensional dataset and including regression of the global signal, a controversial procedure. One should be careful in this context to ensure that the observed effects, especially the generic effects such as reduced segregation between default and executive networks, are not an artifact of this preprocessing pipeline. More information is needed here in order to make this conclusion with more certainty. Here are specific questions and recommendations which the authors could consider and implement to clarify the robustness of their findings:

- a) What is the total motion artifact load for each studied subject?
- b) What is the correlation of the canonical variates with nuisance variables, including subject head motion, and subject total variance described by the global signal?
- d) What happens to the results when the global signal is not regressed?
- c) What happens to the results when the authors exclude high-motion subjects (i.e. only keep low-motion subjects) and implement a less aggressive preprocessing strategy?

3. Other comments

- Figure 7 and Supplementary Figure 5: It was unclear to me whether the authors here have chosen to

focus only on canonical features they have observed in the main results. For instance, in the Gordon parcellation, they have omitted the best performing covariate – is this because it was not significant? In general, the procedure for detection here should be equivalent to that performed in the main analysis.

- Page 11, line 8: “correlations within 0.24 and 0.88”. I wasn’t certain what 0.24 referred to – it seems that all significant correlations exceeded 0.75.

- The online methods should be moved back to the main manuscript, unless there are specific constraints on space limits.

- The paper is well presented. Minor additional suggestions: variable subscripts such as “FDR” should not be italicized. Figure 2b: six identical matrices are apparently shown, rather than six distinct matrices, one for each individual subject.

Reviewer #2 (Remarks to the Author):

What are the major claims of the paper?

Xia et al applied sparse Canonical Correlation Analysis (sCCA) to a large sample (n=663; from the Philadelphia Neurodevelopmental Cohort) to identify linked dimensions of functional connectivity and psychopathology. The authors identified four dimensions, corresponding to psychopathological symptoms primarily related to mood, psychosis, fear, and externalising behavior, each of which was associated with unique and shared patterns of dysconnectivity within and between functional networks. Some significant effects of age and gender were also obtained for connectivity patterns associated with the mood and psychosis dimensions. The linked dimensions were moderately (0.24-0.75) replicable in a hold-out dataset (n=336).

Are they novel and will they be of interest to others in the community and the wider field?

Congruent with the RDOC approach to understanding mental ill-health, this paper describes the novel identification of linked connectivity-psychopathology dimensions in a developmental cohort. It represents an important step in the quest to identify better, developmentally informed biomarkers for key psychiatric dimensions that may ultimately improve our ability to diagnose, to make prognoses, and to treat. As such, it will be of considerable interest to others in the community and the wider field.

Is the work convincing, and if not, what further evidence would be required to strengthen the conclusions?

The paper is a novel and elegant investigation that addresses an important and timely research question – the data are sound, the analyses reflect best practice for the field, the results are compelling, and paper is very well written. A particularly noteworthy feature is that both the data and analysis code are freely available and readily accessible. That said, the data and analyses are presented at a rather high level. As a result, some important details, particularly of some of the results obtained, are glossed over and a consideration of nuance is omitted. I have highlighted some clarifications below that would improve the accessibility of the paper for the reader and strengthen its conclusions. In addition a greater effort could be made to situate the findings within the relatively large body of other outputs yielded by the same dataset (e.g., Shanmugan et al. 2016 and Calkins et al. 2015). Finally, the replicability of some of the findings appears to be overstated and should be presented and discussed in greater detail.

Regarding the sample itself, it is important to note that this is a community sample. What is the

proportion of clinical levels of psychopathology present? Although these details have been very well documented in other publications, a line or two of context for the present study would be beneficial to readers.

The processing pipeline used is highly appropriate and has been shown elsewhere to be successful at minimising motion. It is very helpful that the authors state the precise thresholds used to exclude participants with excessive motion. The precise calculation used to define framewise displacement is not stated, however – I assume from other recent work by the senior author that it's the Jenkinson calculation, but please clarify. Spike regression/censoring does not appear to have been performed – please confirm.

Regarding use of the Power parcellation and community structure, a line or two describing the parcellation (how it was derived) could be added for completeness. Similarly, a line describing the alternative parcellation would be informative. It would also be informative to comment on/add citations regarding the suitability of the Power scheme for the developmental population.

I was a little confused at the fact that, rather than making use of the psychiatric dimensions previously identified using factor analysis of the clinical data (e.g., presented in Shanmugan et al.) and relating those to patterns of functional connectivity (e.g., using MDMR or similar), the analytic approach adopted here (sCCA) generates a new set of dimensions/factors. Could the authors comment on why this approach was chosen? Further, there is generally little consideration of how these new linked dimensions relate to factors previously described or even to the RDOC dimensions – the authors should relate the current set of dimensions to those previously described.

Regarding the observation that only 37 of 111 psychiatric symptoms reliably contributed to at least one of the four dimensions – this suggests that a lot of symptoms are being omitted from these connectivity-guided dimensions. A consideration of the symptoms omitted is important – particularly with regard to the replication analysis (discussed further below).

The description of the results provided on page 8-9 is rather cursory. A more detailed consideration of the results is necessary in order to support some of the claims being made. For example, on page 9, it is stated that, "each canonical variate was comprised of unique patterns of dysconnectivity." However, the summary of the results provided doesn't support that case, and instead suggests there are really only 3-4 patterns that were observed, with a lot of overlap between the dimensions. Specifically, the authors report that elevated connectivity in the default network was associated with both mood and psychosis dimensions, while reduced connectivity in the default network was associated with the fear and externalising dimensions. Elevated connectivity in the in frontoparietal and salience networks was common to the mood and fear dimensions, and reduced segregation between default and executive networks was common to all four dimensions. The more detailed analysis of common dysconnectivity features presented in the second half of page 9 is additionally informative with regard to these shared patterns. However, I am not able to discern what are the unique patterns of dysconnectivity – nor are any highlighted in Figure 4. A more in-depth description of the unique patterns obtained and some way of highlighting them visually is needed. This is particularly important because on page 15 there is a discussion of several results that I don't see presented in the Results section – e.g., enhanced connectivity between the DAN and FP networks and between VAN and salience in mood dimension. Presumably these reflect some of the unique patterns but they do not appear to have been explicated anywhere before that in the manuscript.

Regarding the replication analysis (presented on page 11), it is stated that "sCCA identified four canonical variates that highly resemble the original four linked dimensions." The authors report correlations of loadings between the discovery and replication analysis as low as 0.24, which cannot

be considered indicative of two things that “highly resemble” one another. The full details of the correlations are not presented (i.e., for each of the 4 factors) - we are given the range but not all correlation values. Please present the replication results in full (in the Results section, including a statement of which dimension didn’t replicate). For example, what is the overlap between the specific psychiatric dimensions identified in the two analyses (i.e., overlap in the 37 out of 111)? Did the age and gender findings replicate? These issues, and the moderate evidence for replication of the findings should be addressed in the Discussion section. Finally, I am not really clear what is being presented in Supplementary Figure 8.

Clare Kelly

Reviewer #3 (Remarks to the Author):

I very much enjoyed reading this original paper. The authors should be complimented for their hard work. Replication on an independent sample is clearly one of the strength of the paper, together with the large-sample size and adequate thresholding strategies (which constitute the major drawbacks in a majority of brain imaging research now days). I do have quite a few comments and suggestions which I hope you will find useful for a revision of your paper.

Introduction.

P4. “... complex linear relationships between two-high-dimensional datasets...”:

Even if this is not specific to the sCCA approach presented here, could the authors clarify how the question of more than two classification categories in play has been managed (more than 2 psychopathological dimensions for instance, etc.)?

Linear kernels often provide more interpretable findings than non-linear ones, but I was wondering if there is any other argument in support of using this linear classifier here?

Results-Discussions.

P9. “...The psychosis dimension similarly showed elevated connectivity within the default-mode network and its reduced segregation from executive networks...”:

I would be interested to see these results more broadly discussed in reference to DMN changes in schizophrenia (eg. Lefort-Besnard et al, Hum Brain Mapp 2017). I notably find interesting that the trait approach chosen here seems compatible with what we also know from RSN state changes associated with the presence/absence of psychotic symptoms. You can for instance refer to RSN changes related to the dynamics (ignition, ON-phase, extinction) of paroxistic psychotic symptoms (eg., Lefebvre et al, Hum Brain Mapp 2016).

Methods.

The authors may wish to consider Supplementary Table 1 (sample description) to appear in the main text.

Regarding supplementary Table 3, I noticed that 4 on 6 items for “psychosis” referred to hallucinatory experiences. I feel that the authors should acknowledge somewhere that hallucinations and psychosis are only partially overlapping concepts and that PSY001, PSY029, PSY050 or PSY060 items should at least be associated with one more symptom under the DSM A-criteria (eg. PSY070 or PSY071) to have a clinical significance. The recent editorial from Waters F. et al. Psychol Med 2017.

Minor comment: the authors may wish to consider changing "neurodevelopment" by "development" and "neuropsychiatric" by "psychiatric".

Reviewer #4 (Remarks to the Author):

This paper describes a very interesting and important topic of how individual variation in connectome organization relates to pathophysiology in a group of developing adolescents and adults. My points include:

- * The paper is very well written. As a minor comment: perhaps the introduction could be a little bit shortened.
 - * How was the discovery and replication dataset formed? With the two sets originating from the same dataset, statistically splitting the 2 sets is not a big advantage (or difference).
 - * an in-text description of the 111 clinical items (or a categorization of this) would be very nice. Now 'clinical items' remain a little bit vague.
 - * page 6, last paragraph: 'Of these seven canonical variates, three were 17 significant (Pearson correlation $r = 0.71$, PF DR < 0.001 ; $r = 0.70$, PF DR < 0.001 , $r = 0.68$, PF DR < 0.01 , 18 respectively) (Fig. 2b), with the fourth showing a trend towards significance ($r = 0.68$, PF DR = 0.07, 19 Puncorrected = 0.04).'
- this paragraph was a little bit unclear to me. What was examined here with the 7 variates, and what was 'significant'? What was tested against what and how?
- * Overall, I found the methodology and statistical analysis a little bit hard to follow. I appreciate and understand the complexity of this approach, but explaining it in a 'simple' matter would be very interesting, and make it more clear to the reader which is less advanced in statistical procedures.
 - * The authors describe that the networks were restricted to the top 10 percent of most variable connections. I have a major concern with this, as the most variable connections are potentially also the most 'noisy' connections as they are the least consistent across subjects. Noisy connections can have a strong impact on graph metrics (see the recent work of Zalesky 2016 Neuroimage and Heuvel 2016 Neuroimage). I think it is vital to show that similar results are also found when no 'thresholding' of some sort is performed. Can the authors include such an analysis as validation?
 - * The examination was limited to functional connectivity, and extension of findings to structural connectivity (of which is available in this dataset) would be very interesting.
 - * parcellation: one specific parcellation atlas was chosen, but common practise is to show robustness of findings when other parcellation atlases are used for network reconstruction. Can the authors briefly show findings across other atlases /other resolutions?
 - * the term 'dysconnectivity' was used, but given the notion that the included subjects did not include actual 'patients', perhaps variation in connectivity is a better term? Also in Figure 4 the authors use 'increased' and 'decreased' but perhaps, 'lower'/'higher' is a better term to use in this context?
 - * Figure 2. The plots are very nice, but perhaps a little bit misperceiving. The sCCA is designed to find this type of correlations, so the results presented are the direct consequence of the sCCA analysis.

Perhaps a good null-condition in this context is to shuffle the FC weights across the selected edges in the network, and then re-perform the sCCA analysis. How many significant axis were found in this null-condition, and are the number of sCCA patterns found now significantly exceeding what one would find in the null-condition?

REVIEWER #1

This is a data-driven analysis of a large developmental dataset. The aim is to link symptoms of psychopathology with abnormalities in resting-state functional-connectivity MRI using canonical correlation analysis, a dimensionality reduction technique which maximizes correlations between latent variables. The main observed effect is a discovery of strong relationships between specific functional network abnormalities and 3-4 distinct clusters of symptoms (dimensions of psychopathology).

I was impressed with several aspects of this study which clearly make it stand out from the pack. First, the primary dataset is big (n=663) and quite unique in its combination of large size, a detailed psychopathological assessment and resting-state imaging. Second, the authors employed statistically principled machine learning analysis techniques, rather than the more hand-wavy tools of network neuroscience. Third, the authors have taken care to ensure that their findings are robust, with permutation tests, resampling, and most importantly replication of the main effects in an independent part of their dataset (n=336); this is excellent practice and should be compulsory for the field more generally... The paper is well presented.

We thank Reviewer #1 for the positive appraisal of our work, and are happy to integrate the thoughtful feedback offered.

1. First, I feel that the study is presented in quite a convoluted way. The main results are essentially about the mapping of symptoms to circuits. This is valuable, but cannot in itself allow us to make progress towards a biologically-grounded understanding of psychiatric disorders, despite what the study claims.

To appreciate the main conceptual concern, consider how the present abstract might be rewritten from the perspective of a more biologically-grounded field, such as respiratory medicine:

“[a] Physiological abnormalities associated with respiratory disorders do not map well to existing diagnostic categories. High co-morbidity and overlapping symptom domains suggest abnormalities that cut across clinical diagnoses. [b] Here we sought to identify respiratory-tract based dimensions of respiratory pathology. We found that several dimensions of respiratory pathology -- rhinorrhea, cough and wheeze -- were highly associated with distinct patterns of pathophysiology. [c] These results delineate physiologically guided dimensions of respiratory pathology that cut across traditional diagnostic categories, which could serve as a foundation for developing objective biomarkers in respiratory medicine.”

Consider the problems inherent in these statements. Statement [a] is presented as a foundational limitation that is holding back progress for objective diagnosis. Yet in respiratory medicine no one seems concerned by the fact that the symptom of a runny nose (rhinorrhea) cuts across clinical diagnostic categories; it does so by design, as an end-stage process of distinct upstream mechanisms (diseases). [b] the present study maps psychiatric symptoms to abnormalities of neural circuits, much in the same way as rhinorrhea could be mapped to an inflamed nasal mucosa. This is a valuable contribution as such mappings are largely absent in psychiatry. But it should not be surprising that these clusters of symptoms cut across diagnostic categories; in fact, it is expected that they should do so! [c] implies that such mappings could serve to develop objective biomarkers of psychiatric disorders, which is arguably not the case here: understanding an association between inflammation and rhinorrhea cannot delineate

diseases which give rise to the inflammation. The same seems to hold for psychiatric symptoms and circuit abnormalities.

Put another way, the present study is not a study of psychiatric disorders (upstream), but a study of psychiatric symptoms (downstream). This becomes apparent when the authors find that “features selected by multivariate analyses generally accord with clinical phenomenology”. In contrast, consider a related work on biotypes (Clementz et al., 2015) which takes a more upstream view and identifies biomarkers independently of clinical features, albeit in a more restricted disease setting of psychosis. It seems to me that such an upstream approach is more explicitly trying to achieve what the authors set out to do here, namely to classify psychiatric disorders in an objective and quantitative manner.

This distinction is critical, and is a major distinguishing feature of RDoC (Cuthbert and Insel, 2013). “The distinctions between RDoC and the DSM and ICD systems can be captured by seven major points that include both conceptual and practical differences. First, the approach incorporates a strong translational research perspective. ***Rather than starting with symptom-based definitions of disorders and working toward their pathophysiology, RDoC inverts this process. Basic science - in genetics, other areas of neuroscience and behavioral science - serves as the starting point, and disorders are considered in terms of disruptions of the normal-range operation of these systems, with an emphasis on the mechanisms that serve to result in dysfunctions of varying degrees***.”

In this context, my suggestion to the authors is to rewrite major portions of the text (in the abstract, introduction, results and discussion) which concern the motivation and the interpretation of their work. They should more clearly motivate and interpret their contributions, by de-emphasizing aspects concerning classification of psychiatric disorders and emphasizing and motivating the rationale for linking symptom groupings with circuit abnormalities.

We appreciate this important feedback. We would like to clarify that the paper is indeed motivated by the fact that current symptom-based diagnostic labels classifying psychiatric disorders do not “carve nature at its joints”, giving rise to marked heterogeneity and co-morbidity. While it is certainly informative and important to study “biotypes” within symptom categories as pointed out by the reviewer (Clementz, et al., *Am. J. Psychiatry*, 2015 and Drysdale et al., *Nature Medicine*, 2017, both cited in the manuscript), here we pursue a complementary approach – using functional connectivity to guide symptom groupings that are *trans*-diagnostic, so they are not bound by our current discrete classifications. We would like to emphasize that this approach is not in tension with RDoC, as we aim to delineate the circuit-level abnormalities that may underlie dimensions of psychopathology that cross the typical diagnostic categories provided by the DSM. Nonetheless, we incorporated the thoughtful feedback from the reviewer, and have made changes to the manuscript in an effort to clarify the limitations of this approach and distinguish it from prior work using biotypes. Specifically, we now note in the revised Introduction:

It should be noted that the approach of the current study is distinct from prior work discovering biotypes *within* categories of psychopathology, based purely on imaging features themselves (e.g. functional connectivity (Drysdale, *Nature Medicine*, 2016) and gray matter density (Clementz et al, *AJP*, 2016)). In contrast, we seek to link a broad range of symptoms that are present *across* categories to individual differences in functional brain networks. Such an approach has been successfully applied in prior work

on neurodegenerative diseases (Avants, *Neuroimage*, 2014) as well as normal brain-behavior relationships (Smith et al, *Nature Neuroscience*, 2015).

Furthermore, we now re-emphasize this point in the revised Limitations:

It should be emphasized that our approach did not seek to define biotypes *within* clinical diagnostic categories in a fully data-driven manner, as in influential prior work (Clementz et al, *AJP*, 2016; Drysdale, *Nature Medicine*, 2016). Rather, here we sought to provide complementary understanding of heterogeneity by linking symptoms that are present *across* clinical diagnostic categories to alterations of functional connectivity, uncovering dimensions of psychopathology that are guided by and linked to underlying network abnormalities. However, this approach necessarily is limited by the clinical data being used, in this case item-level data from a structured clinical interview.

2. The processing of functional MRI data is a field fraught with dangers, including global signal regression, motion correction, and vigilance control. The authors have pursued an aggressive strategy to remove sources of artifact from their data, but it is important to show that this strategy has not led to an undue deformation of the (already) fairly low-dimensional signal and the resulting description of their functional networks.

We share the reviewer's sentiment that methodological choices regarding fMRI preprocessing are of critical importance. Indeed, our group was among the first to identify the impact of motion artifact on functional connectivity (Satterthwaite et al., *Neuroimage* 2012), how pre-processing can be used to mitigate motion artifact (Satterthwaite et al., *Neuroimage* 2013a), and how motion may systematically bias estimates of brain development (Satterthwaite et al., *Neuroimage* 2013b). More recently, we systematically evaluated a wide range of confound regression methods for mitigating the impact of motion artifact on functional connectivity using a broad range of benchmarks (Circic et al., *Neuroimage* 2017; also see Satterthwaite et al., *Human Brain Mapping* 2017 for a review). Notably, in Circic et al. (2017), we demonstrated that global signal regression coupled with appropriate censoring methods can markedly attenuate motion artifact while conserving network signal of interest. Indeed, while the field has not reached complete consensus, recent analyses and reviews tend to support the use of global signal regression (Burgess et al., *Brain Connectivity* 2017; Power et al., *Neuroimage* 2017; Power et al., *Trends Cog Sci* 2017; Byrge and Kennedy, *Neuroimage* 2017; Murphy et al., *Neuroimage* 2016).

a) What is the total motion artifact load for each studied subject?

As detailed in the new **Supplementary Figure 2** shown below, we used stringent exclusion criteria to remove high-motion participants. Specifically, as in prior work, we removed participants with mean framewise displacement (FD) of >0.2 mm, as well as participants who had more than 20 volumes with a FD of >0.1mm. Using these criteria, of the n=1405 participants for whom resting-state functional connectivity data was available, n=229 were removed due to excessive motion

(see Panel A, below). After applying other exclusion criteria (as detailed in **Supplementary Figure 1**), 999 subjects were included in the final analysis. The histogram of mean FD for the original (n=1405), combined (n=999), discovery (n=663), and replication (n=339) samples are displayed below.

Supplementary Figure 2 | In-scanner motion of subjects. (a) 1405 out of 1601 participants of PNC had acquired resting-state fMRI. The histogram shows the distribution of mean framewise displacement using the Jenkinson calculation. The exclusion criteria of motion for the final sample is 0.2mm or greater, which is colored in red (n=229). (b) After applying all exclusion criteria, including health, structural and functional imaging quality exclusion criteria, 999 subjects were included in the final sample. The histogram shows the head motion distribution of the final sample, which consists of a discovery sample (c), and a replication sample (d).

b) What is the correlation of the canonical variates with nuisance variables, including subject head motion, and subject total variance described by the global signal?

Anticipating the risk of motion driving correlations between functional connectivity and psychopathology, as part of the pre-processing steps detailed in the original manuscript we regressed motion out of edge-wise connectivity before we ran CCA analysis. Here, to further demonstrate that this procedure was successful, we now display the relationship between the four canonical modes of covariation and subject motion. As expected, there was no significant relationship between motion and either clinical (Panel A) or brain scores (Panel B).

While somewhat beyond the scope of the present paper, in prior methodological papers using data from the PNC, we have examined how much variance is explained by confound regressors such as GSR (see Satterthwaite et. al, *Neuroimage* 2013). In that study, we found that three parameters (including global signal, white matter and CSF) accounted for 46% of total signal variance (Figure 8 in the original paper, reproduced below).

Control analysis investigating the effect of including increasing numbers of noise regressors on regression diagnostic outcomes and connectivity-based outcome measures. Whole-brain confound regression models are shown in black. The gray line plot indicates confound regression models that included a variable number of noise regressors. All noise regression models consist of 3 confound regressors that were constructed from real data (global signal, white matter, CSF), plus a variable number (6, 15, 33) of randomly generated noise regressors. Adding noise regressors does not increase the variance explained by the model as measured by the adjusted r^2 (A). Data from Figure 8 in Satterthwaite et al. *Neuroimage* 2013.

Furthermore, this has been investigated in comprehensive detail across multiple datasets in a recently published paper by Power et al (*Neuroimage*, 2017).

c) What happens to the results when the global signal is not regressed?

We are happy to clarify our choice of pre-processing pipeline. As noted above, we chose to use a pipeline that includes GSR based on the results of extensive benchmarking experiments. In a recent paper (Ciric et al., *Neuroimage*, 2017) we compared 14 common pipelines using a wide variety of benchmarks, including the residual relationship between motion and connectivity, distance-dependence of motion artifact, and sub-network identifiability. These experiments revealed that GSR was highly effective at mitigating the relationship between motion and connectivity. Specifically, less than 1% of network edges had a significant relationship between motion and connectivity using this pipeline. In contrast, when using a similar pipeline that lacked GSR (i.e., the commonly used 24P pipeline), **79%** of edges had a significant relationship between motion and connectivity (see **Figure 2** from Ciric et al., *Neuroimage*, 2017 reproduced below). Critically, pipelines such as this one that were more “aggressive” in their removal of motion artifact in fact tended to be *better* at identifying network modules, indicating that de-noising did not remove valuable signal from the data (see **Figures 5 and 6** in Ciric et al., *Neuroimage*, 2017). Based on these previous experiments, we have retained our current pre-processing pipeline as it provides both excellent control of the confounding influence of motion and also aids in identification of network topology.

Number of edges significantly related to motion after de-noising. Successful de-noising strategies reduced the relationship between connectivity and motion. The number of edges for which this relationship persists provides evidence of a pipeline's efficacy. A, The percentage of edges significantly related to motion in a 264-node network defined by Power et al. (2011). Fewer significant edges is indicative of better performance. B, The percentage of edges significantly related to motion in a second, 333-node network defined by Gordon et al. (2016). C, Renderings of significant edges with QC-FC correlations of at least 0.2 for each de-noising strategy, ranked according to efficacy. Strategies that include regression of the mean global signal are framed in blue and consistently ranked as the best performers. **Figure 2** From Ciric et al., *Neuroimage* 2017.

d) What happens to the results when the authors exclude high-motion subjects (i.e. only keep low-motion subjects) and implement a less aggressive preprocessing strategy?

As described in the methods section and displayed in **Supplemental Figure 1** and **Supplemental Figure 2**, we have excluded high-motion subjects from our analysis, including those that had an average framewise motion more than 0.2 mm or more than 20 frames exhibiting movement in excess of 0.25 mm ($n=229$). Also see response to Comment 2a, above.

3. Figure 7 and Supplementary Figure 5: It was unclear to me whether the authors here have chosen to focus only on canonical features they have observed in the main results.

We appreciate the feedback, and are happy to clarify that the same procedure from the discovery sample was followed in the replication sample, except that no parameter tuning was performed in the replication sample. Specifically, a scree-plot was first used to examine the covariance explained by each canonical mode (Figure 7a, and the left column of **Supplementary Figure 5**). We used the elbow of the scree plot to choose the first four modes for follow-up analyses (box in **Figure 7a**). As in the discovery sample, we performed a permutation test to ascertain if the correlation between connectivity and clinical features in these four modes exceeded that of the null distribution. Three of the four modes were significant after FDR-correction. As in the discovery sample, we named these significant modes based on the clinical item loadings, as well as on the correspondence in clinical loadings with the discovery sample (see next comment).

Figure 8 | Linked dimensions of psychopathology were replicated in an independent sample. All procedures were repeated in an independent replication sample of 336 participants. **(a)** The first four canonical variates in the replication sample were selected based on covariance explained for further identical analysis as in discovery steps. Dashed line marks the average covariance explained. **(b)** The mood, fear, and externalizing behavior dimensions were significant by permutation testing. Corresponding variates are circled in (a) Error bars denote standard error. $**P_{FDR} < 0.01$.

4. Page 11, line 8: “correlations within 0.24 and 0.88”. I wasn’t certain what 0.24 referred to – it seems that all significant correlations exceeded 0.75

We agree that this was unclear. These are the correlations of the loadings of the clinical items between the two samples. We have revised this statement as follows:

Although this replication sample was half the size of our original discovery sample, sCCA identified four canonical variates that highly resemble the original four linked dimensions of psychopathology. Specifically, the correlations between the clinical loadings in the discovery samples and those in the replication samples were $r=0.85$ for psychosis ($P_{FDR} = 4.4e-16$), $r=0.73$ for externalizing ($P_{FDR} = 4.4e-16$), $r=0.59$ for fear ($P_{FDR} = 8.43e-12$), and $r=0.23$ for mood ($P_{FDR} = 0.01$).

5. *The online methods should be moved back to the main manuscript, unless there are specific constraints on space limits.*

While we completely agree, the journal style of *Nature Communications* only allows for main text of 5,000 words. This unfortunately precludes us from including more of the methods in the main text, as we would prefer. Nonetheless, we have now tried to integrate greater methodological detail into the manuscript where appropriate and space allows. Several specific examples are detailed below:

Page 6: As features that do not vary across subjects cannot be predictive of individual differences, we limited our analysis of connectivity data to the top 10 percent most variable connections, as measured by median absolute deviation, which is more robust against outliers than standard deviation.

Page 7: Using elastic net regularization ($L1 + L2$) and parameter tuning over both the clinical and connectivity features, sCCA was able to obtain a sparse and interpretable model while minimizing over-fitting.

Page 7: Significance of each of these linked dimensions of symptoms and connectivity was assessed using a permutation test, which compares the canonical correlate of each variate to a null distribution built by randomly re-assigning subjects' brain and clinical features.

6. *Variable subscripts such as "FDR" should not be italicized.*

We have changed this formatting as suggested.

7. *Figure 2b: six identical matrices are apparently shown, rather than six distinct matrices, one for each individual subject.*

We have updated this figure as appropriately suggested.

REVIEWER #2

Xia et al applied sparse Canonical Correlation Analysis (sCCA) to a large sample (n=663; from the Philadelphia Neurodevelopmental Cohort) to identify linked dimensions of functional connectivity and psychopathology. The authors identified four dimensions, corresponding to psychopathological symptoms primarily related to mood, psychosis, fear, and externalizing behavior, each of which was associated with unique and shared patterns of dysconnectivity within and between functional networks. Some significant effects of age and gender were also obtained for connectivity patterns associated with the mood and psychosis dimensions. The linked dimensions were moderately (0.24-0.75) replicable in a hold-out dataset (n=336).

Congruent with the RDOC approach to understanding mental ill-health, this paper describes the novel identification of linked connectivity-psychopathology dimensions in

a developmental cohort. It represents an important step in the quest to identify better, developmentally informed biomarkers for key psychiatric dimensions that may ultimately improve our ability to diagnose, to make prognoses, and to treat. As such, it will be of considerable interest to others in the community and the wider field.

The paper is a novel and elegant investigation that addresses an important and timely research question – the data are sound, the analyses reflect best practice for the field, the results are compelling, and paper is very well written. A particularly noteworthy feature is that both the data and analysis code are freely available and readily accessible.

We thank Reviewer #2 for the positive appraisal of the manuscript and appreciate the useful feedback, which we have integrated as detailed below.

1. Regarding the sample itself, it is important to note that this is a community sample. What is the proportion of clinical levels of psychopathology present? Although these details have been very well documented in other publications, a line of two of context for the present study would be beneficial to readers.

We agree, and now specify this information explicitly. Furthermore, in the revised manuscript we provide additional detail regarding the prevalence of screening-level diagnoses in this sample in new **Supplementary Table 2**.

Participants in the PNC were recruited from Children’s Hospital of Philadelphia pediatric network in the greater Philadelphia area. The sample includes n=1,601 youth ages 8-22 years old (551 female; see **Figure 1**). In this community-based study, participants were not recruited from psychiatric services. As such, the prevalence of screening into specific psychopathology categories generally aligned with epidemiologically ascertained samples, as previously described (Calkins et al., *J. Child Psychology & Psychiatry*, 2015; see **Supplementary Table 2**).

Supplementary Table 2: Clinical Psychopathology Levels in the PNC

Psychopathology Categories	All sample (n=1601)	Total analyzed sample (n=999)	Discovery (n=663)	Replication (n=336)
Mania	1.2%	1.3%	1.1%	1.8%
Depression	13.2%	15.3%	16.3%	13.4%
Bulimia	0.4%	0.3%	0.5%	0.0%
Anorexia	1.0%	1.4%	1.8%	0.6%
Generalized Anxiety Disorder	1.7%	1.6%	1.8%	1.2%
Separation Anxiety Disorder	4.7%	3.9%	4.1%	3.6%
Social Phobia	23.1%	24.8%	25.2%	24.1%
Panic Disorder	1.0%	0.8%	1.1%	0.3%

Agoraphobia	5.7%	5.8%	6.5%	4.5%
Obsessive-Compulsive Disorder	2.8%	2.7%	2.6%	3.0%
Post-Traumatic Stress Disorder	11.7%	12.4%	12.4%	12.5%
Psychosis	6.7%	6.6%	7.5%	4.8%
Attention-Deficit Disorder	17.3%	15.3%	15.2%	15.5%
Conduct Disorder	8.9%	8.6%	7.8%	10.1%

2. *The processing pipeline used is highly appropriate and has been shown elsewhere to be successful at minimising motion. It is very helpful that the authors state the precise thresholds used to exclude participants with excessive motion. The precise calculation used to define framewise displacement is not stated, however – I assume from other recent work by the senior author that it's the Jenkinson calculation, but please clarify. Spike regression/censoring does not appear to have been performed – please confirm.*

We are happy to clarify this important point. Indeed, we used the Jenkinson calculation for FD. We now note this in the revised methods. Furthermore, as in prior papers in this dataset, we used a 36-parameter confound regression model with despiking as described in the *Functional Preprocessing* subsection in the Online Methods.

3. *Regarding use of the Power parcellation and community structure, a line or two describing the parcellation (how it was derived) could be added for completeness. Similarly, a line describing the alternative parcellation would be informative. It would also be informative to comment on/add citations regarding the suitability of the Power scheme for the developmental population.*

We agree, and have added the descriptions of the Power and Gordon parcellation to the Online Methods:

We built a functional connectivity network using the residual timeseries (following denoising) of all parcels of a common parcellation. The parcellation used in the main analysis consists of 264 spherical nodes of 20 mm diameter distributed across the brain (Power et al., *Neuron*, 2011). The *a priori* communities for this set of nodes were originally delineated using the Infomap algorithm (Rosvall and Bergstrom, *PNAS*, 2008) and were replicated in an independent sample. This parcellation was particularly suitable for our analysis as it has been previously used for studying developmental changes in connectivity and network modularity (Power et al, *Neuron* 2010) and has been used as part of several studies in this dataset in the past (Gu et al., *PNAS*, 2015; Satterthwaite et al., *Cerebral Cortex*, 2015; Ciric, et al., *Neuroimage*, 2017; Chai et al., *Network Neuroscience*, 2017). As part of the supplementary analysis to demonstrate the robustness of the results independent of parcellation choices (**Supplementary Figure 6**), we also

constructed networks based on an alternative parcellation developed by Gordon et al. (Cerebral Cortex, 2016). This set of nodes was derived using edge detection and boundary mapping to define areal parcels (Gordon et al., Cerebral Cortex, 2016).

4. A greater effort could be made to situate the findings within the relatively large body of other outputs yielded by the same dataset (e.g., Shanmugan et al. 2016 and Calkins et al. 2015). I was a little confused at the fact that, rather than making use of the psychiatric dimensions previously identified using factor analysis of the clinical data (e.g., presented in Shanmugan et al.) and relating those to patterns of functional connectivity (e.g., using MDMR or similar), the analytic approach adopted here (sCCA) generates a new set of dimensions/factors. Could the authors comment on why this approach was chosen? Further, there is generally little consideration of how these new linked dimensions relate to factors previously described or even to the RDoC dimensions – the authors should relate the current set of dimensions to those previously described.

We agree, and are happy to clarify. In prior work, we used factor analyses of clinical instruments to delineate dimensions of psychopathology. This prior work included both a standard correlated-traits model (Calkins et al. *J. Child Psychology & Psychiatry* 2015) as well as a bifactor model that incorporated overall psychopathology (Shanmugan et al. *Am J. Psychiatry* 2016; Kazkurkin et al., *Biological Psychiatry*, 2016). In the present study, we were explicitly interested in finding clinical dimensions of psychopathology that were strongly and specifically linked to individual differences in functional connectivity. The critical difference between these approaches is that dimensions derived by our prior work were driven solely by covariance in the clinical symptomatology, whereas the dimensions delineated here were guided by associations with functional connectivity. We now make this point explicitly in the revised introduction:

Despite the increasing interest in describing how abnormalities of brain network development lead to the emergence of psychiatric disorders, existing studies have been limited in several respects. First, most have adopted a categorical case-control approach, or only examined a single dimension of psychopathology (e.g. Satterthwaite et al., *Molecular Psychiatry*, 2015), and are therefore unable to capture heterogeneity across diagnoses. Second, dimensional psychopathology derived from factor analyses, including our prior work (Calkins et al. *J. Child Psychology & Psychiatry* 2015; Shanmugan et al. *Am J. Psychiatry* 2016; Kazkurkin et al., *Biological Psychiatry*, 2016), were solely driven by covariance in the clinical symptomatology, rather than being guided by both brain and behavior features. Third, especially in contrast to adult studies, existing work in youth has often used relatively small samples (e.g., dozens of participants). While multivariate techniques allow the examination of both multiple brain systems and clinical dimensions simultaneously, such techniques usually require large samples.

Nonetheless, we agree that it would be informative to understand how similar these connectivity-guided dimensions are to those derived from pure clinical items.

Accordingly, we now include additional supplementary analyses where we relate the clinical scores from our analysis to scores previously derived from the bifactor model (new **Supplementary Figure 9**). As shown below, the clinical dimensions derived from connectivity-guided sCCA analysis are significantly correlated with, but not identical to, those derived from the bifactor model (Shanmugan et al. *Am J. Psychiatry* 2016). In particular, as expected in an approach that does not individually model overall psychopathology, all dimensions show a substantial correlation ($r \sim 0.5$ or greater) with the overall psychopathology factor from the bifactor model described in Shanmugan et al. (2016). Furthermore, sub-factors from the bifactor model were variably related to connectivity-guided dimensions from sCCA (range: $r = 0.17$ to 0.71). We now note the relationships with prior factor models explicitly in the revised Results:

Results, page 7:

The connectivity-guided clinical dimensions were significantly correlated with, but not identical to, those previously obtained from (Shanmugan et al. *Am J. Psychiatry* 2016), see **Supplementary Figure 10**).

A
Overall Psychopathology vs. Canonical Variates

B
Bifactor Model vs. Canonical Variates

Supplementary Figure 10 | Correlations between canonical variates and previous factor analysis model. To understand how similar connectivity-guided dimensions of psychopathology are to those derived from pure clinical items, we examined the correlation between the canonical variate clinical scores and **(a)** overall psychopathology score, and **(b)** dimensional bifactor models scores, both initially reported in Shanmugan et al., Am. J. Psychiatry, 2016.

5. The description of the results provided on page 8-9 is rather cursory. A more detailed consideration of the results is necessary in order to support some of the claims being made. For example, on page 9, it is stated that, “each canonical variate was comprised of unique patterns of dysconnectivity.” However, the summary of the results provided doesn’t support that case, and instead suggests there are really only 3-4 patterns that were observed, with a lot of overlap between the dimensions. Specifically, the authors report that elevated connectivity in the default network was associated with both mood and psychosis dimensions, while reduced connectivity in the default network was associated with the fear and externalising dimensions. Elevated connectivity in the in frontoparietal and salience networks was common to the mood and fear dimensions, and reduced segregation between default and executive networks was common to all four dimensions. The more detailed analysis of common dysconnectivity features presented in the second half of page 9 is additionally informative with regard to these shared patterns. However, I am not able to discern what are the unique patterns of dysconnectivity – nor are any highlighted in Figure 4. A more in-depth description of the unique patterns obtained and some way of highlighting them visually is needed. This is particularly important because on page 15 there is a discussion of several results that I don’t see presented in the Results section – e.g., enhanced connectivity between the DAN and FP networks and between VAN and salience in mood dimension. Presumably

these reflect some of the unique patterns but they do not appear to have been explicated anywhere before that in the manuscript.

This is valuable feedback. As suggested, we have now improved the description of the specific individual differences in connectivity in each dimension in the revised Results as detailed below.

Results Page 8:

Specifically, we examined patterns of both *within*-network and *between*-network connectivity (**Supplementary Fig.8**; Online Methods), as this framework has been useful in prior investigations of both brain development (Satterthwaite et al., Neuroimage, 2012; Park et al., Science, 2013) and psychopathology (Sharp et al., Nat Rev Neurol, 2014; Sylvester et al., Trends in Neurosciences, 2012; Kaiser et al., JAMA Psychiatry, 2016). This procedure revealed specific patterns of network-level connectivity that were related to the four dimensions of psychopathology (**Fig.5**). For example, the mood dimension was characterized by a marked increase in connectivity between the ventral attention and salience networks (**Fig.5 a, e, i**), while the psychosis dimension received the highest loadings in connectivity between the default mode and executive systems (salience and fronto-parietal networks (**Fig.5 b, f, j**)). In contrast, increased within-network connectivity of the fronto-parietal network was most evident in the fear dimension (**Fig.5 c, g, k**). Alterations of the salience system were particularly prominent for the externalizing behavior dimension, including lower connectivity with the default mode system and greater connectivity with the fronto-parietal control network (**Fig.5 d, h, l**). Quantitatively, the specific loadings of *within*- and *between*-network connectivity in each dimension did not significantly correlate with each other (all $p > 0.05$), demonstrating that each dimension of psychopathology was characterized by a unique pattern of network dysconnectivity.

6. Regarding the replication analysis (presented on page 11), it is stated that “sCCA identified four canonical variates that highly resemble the original four linked dimensions.” The authors report correlations of loadings between the discovery and replication analysis as low as 0.24, which cannot be considered indicative of two things that “highly resemble” one another. The full details of the correlations are not presented (i.e., for each of the 4 factors) - we are given the range but not all correlation values. Please present the replication results in full (in the Results section, including a statement of which dimension didn’t replicate). For example, what is the overlap between the specific psychiatric dimensions identified in the two analyses (i.e., overlap in the 37 out of 111)? Did the age and gender findings replicate? These issues, and the moderate evidence for replication of the findings should be addressed in the Discussion section. Finally, I am not really clear what is being presented in Supplementary Figure 8.

We greatly appreciate this feedback. As appropriately suggested by the reviewer, we now provide additional details regarding the replication analysis, including correlations

among clinical loadings for each dimension, as well as overlap among clinical items identified by the bootstrap re-sampling analysis. Specifically, these details are now provided in the revised Results on page 11:

Although this replication sample was half the size of our original discovery sample, sCCA identified four canonical variates that highly resemble the original four linked dimensions of psychopathology. Specifically, the correlations between the clinical loadings in the discovery sample and those in the replication sample were $r=0.85$ for psychosis ($P_{FDR} = 4.4e-16$), $r=0.73$ for externalizing ($P_{FDR} = 4.4e-16$), $r=0.59$ for fear ($P_{FDR} = 8.43e-12$), and $r=0.23$ for mood ($P_{FDR} = 0.01$). While the bootstrap analysis identified 37 out of 111 symptoms in the discovery sample to consistently contribute to the four linked-dimensions (**Figure 4**), the same analysis in the replication sample yielded similar sets of symptoms (80%, 64%, 63%, and 50% overlapping for psychosis, externalizing behavior, fear, and mood respectively.)

Furthermore, we now report age and sex effects in the replication sample as well, which provided convergent results with the discovery sample.

Additionally, connectivity patterns associated with mood symptoms increased significantly with age ($P_{FDR} = 0.0082$), while connectivity patterns associated with psychosis symptoms showed a trend towards increasing with age ($P_{uncorrected} = 0.027$, $P_{FDR} = 0.053$). As in the discovery sample, connectivity patterns associated with fear ($P_{FDR} = 0.039$) and mood ($P_{FDR} = 0.0083$) were both elevated in females in the replication sample.

Finally, **Supplementary Figure 8** is the visualization for the identical analysis for the replication sample as the main Figure 2c-f for the discovery sample. This figure is composed of four scatter plots, one for each of the first four canonical variates identified in the replication sample. Each dot represents an individual and is colored by the level of the most heavily weighted clinical item. We now clarify this in the revised legend for this figure.

Supplementary Figure 9 | Linked dimensions in the replication sample accord with those found in the discovery sample. Scatter plots of the first four canonical variates identified in the replication sample, with each dot (represents an individual) colored by the level of the most heavily weighted clinical item in that dimension. Each insert displays the null distribution of sCCA correlation by permutation testing. Dashed line marks the actual correlation. These scatter plots are the counterpart figure of the replication sample, which followed the identical analysis, to Figure 2c-f in the main text of the discovery sample.

REVIEWER #3

I very much enjoyed reading this original paper. The authors should be complimented for their hard work. Replication on an independent sample is clearly one of the strength of the paper, together with the large-sample size and adequate thresholding strategies (which constitute the major drawbacks in a majority of brain imaging research now days).

We appreciate the reviewer's positive appraisal of the study. As detailed below, we are very happy to integrate the helpful feedback.

1. Introduction. P4. "... complex linear relationships between two-high-dimensional datasets...": Even if this is not specific to the sCCA approach presented here, could the authors clarify how the question of more than two classification categories in play has been managed (more than 2 psychopathological dimensions for instance, etc.)?

We are happy to clarify. Traditional methods such as case-control classification (i.e. two-class), or multi-class classification would model the data such that each subject or observation is categorized into one discrete class. These supervised learning techniques, such as support vector machines or random forests, have been fruitful in understanding the neural correlates within established categories of psychopathology. However, our study is motivated by the notion that there are inherent limitations in using discrete diagnostic categories (such as those provided by the DSM) to study heterogeneity of psychopathology. Accordingly, here we used an unsupervised learning technique — sCCA— in order to delineate continuous dimensions of psychopathology. Instead of assigning an individual to a specific class, it allows each dimension to be present in an individual to a varying degree. We believe that this dimensional, trans-diagnostic approach is complementary to traditional classification approaches as it offers a formal way to test whether a mixture of brain circuit abnormalities emerge together in an individual to produce a specific combination of psychiatric symptoms. We have attempted to clarify the motivation of our approach in the revised discussion:

Discussion, Page 13:

In contrast, in this study we used a multivariate analysis technique – sCCA – that allowed simultaneous consideration of clinical and functional connectivity data in a large sample with diverse psychopathology. This method allowed us to uncover linked dimensions of psychopathology and connectivity that cross diagnostic categories yet remain clinical interpretable. Comparing to supervised classification methods (e.g. case-control, or multi-class), where each subject is categorized into one discrete class, as an unsupervised learning technique, sCCA overcomes the inherent limitation of using discrete diagnostic categories (such as those provided by the Diagnostic and Statistical Manual of Mental Disorders (5th edition, APA, 2013)) and allows continuous dimensions of

psychopathology to be present in an individual to a varying degree. In addition, in contrast to “one-view” multivariate studies (such as factor analysis of clinical data or clustering of imaging data), the sCCA-derived clinical dimensions were explicitly selected on the basis of co-varying signals that were present as both individual differences of connectivity and clinical symptoms. Such an unsupervised “two-view” approach has been successfully applied in studies of neurodegenerative diseases and normal brain-behavior relationships. In this dimensional, trans-diagnostic approach, the psychopathology of an individual is represented as a mixture of dimensional brain circuit abnormalities, which together produce a specific combination of psychiatric symptoms.

2. Introduction. Linear kernels often provide more interpretable findings than non-linear ones, but I was wondering if there is any other argument in support of using this linear classifier here?

We completely agree with the reviewer that while non-linear kernels could potentially better model the data, the resulting model is less likely to be interpretable. In addition, we chose a linear kernel in the current study due to the sample size available, as a non-linear and more complex model would naturally have more parameters to estimate and would require a larger sample size to achieve a stable estimation of these parameters. To further enhance the interpretability of the results, we utilized a regularized version of the linear kernel to achieve a sparse representation of both the connectivity and clinical features. We now note this in the revised Online Methods:

Methods, Page 27, sCCA section:

We chose a linear kernel over non-linear implementations of sCCA for two reasons. First, while a more complex model may potentially better fit the data, increased model complexity often results in reduced interpretability. Secondly, a non-linear model may require a larger sample size to accurately estimate the increased number of parameters.

3. Results-Discussion. P9. “...The psychosis dimension similarly showed elevated connectivity within the default-mode network and its reduced segregation from executive networks...”: I would be interested to see these results more broadly discussed in reference to DMN changes in schizophrenia (eg. Lefort-Besnard et al, Hum Brain Mapp 2017). I notably find interesting that the trait approach chosen here seems compatible with what we also know from RSN state changes associated with the presence/absence of psychotic symptoms. You can for instance refer to RSN changes related to the dynamics (ignition, ON-phase, extinction) of paroxysmic psychotic symptoms (eg., Lefebvre et al, Hum Brain Mapp 2016).

We appreciate the reviewer’s feedback and comments. As suggested, the revised discussion now addresses the relevant literature on psychosis and resting-state functional networks, to the degree that length limitations of the journal format allow:

Page 16: Using a purely data-driven analysis, our results support the possibility that loss of segregation between the default mode and executive networks may be a common neurobiological mechanism underlying vulnerability to a wide range of psychiatric symptoms, lending new evidence for the triple-network model of psychiatric disorders (Menon et al, Trends Cogn Sci, 2011; Lefebvre et al, Hum Brain Mapp 2016).

Page 17: Finally, the psychosis dimension exhibited stronger connectivity in default mode network and reduced segregation from executive networks (fronto-parietal and salience). Notably, while prior studies have focused on the central role of default mode dysconnectivity in schizophrenia (Whitefield-Gabrieli and Ford, Annu. Rev. Clin. Psychol, 2012) with mixed evidence for hyperactivity (Zhou et al., Schizophrenia Research, 2007) and hypo-connectivity (Pankow et al., Schizophrenia Research, 2015), the effect within default mode network itself was not nearly as strong as its coupling with the executive networks. Indeed, this finding is consistent with recent data that in psychosis the disruption of connectivity between the default mode and task positive networks is a more consistent feature than dysconnectivity within the default mode itself (Lefort-Besnard et al, Hum Brain Mapp 2017).

Methods

4. The authors may wish to consider Supplementary Table 1 (sample description) to appear in the main text.

We have now moved the Supplementary Table 1 to the main text as part of the new **Figure 1**, as suggested by the reviewer.

5. Regarding supplementary Table 3, I noticed that 4 on 6 items for “psychosis” referred to hallucinatory experiences. I feel that the authors should acknowledge somewhere that hallucinations and psychosis are only partially overlapping concepts and that PSY001, PSY029, PSY050 or PSY060 items should at least be associated with one more symptom under the DSM A-criteria (eg. PSY070 or PSY071) to have a clinical significance. The recent editorial from Waters F. et al. Psychol Med 2017.

We are happy to clarify. Indeed, these specific items regarding perceptual disturbances were found to have significant loadings on the psychosis dimension. However, in addition to these items, other items in the assessment (drawn originally from the SIPS) that measure other domains of psychosis also had significant loadings in this dimension. Specifically, there were significant loadings in the psychosis dimension for disorganized thoughts (SIPS003) and delusions (SIP007, SIP011). To help clarify this, we have now moved the SIPS section right next to the PSY section in the revised **Supplementary Table 3**, and highlighted the items that had significant loadings.

6. *The authors may wish to consider changing “neurodevelopment” by “development” and “neuropsychiatric” by “psychiatric”.*

We have changed this throughout the manuscript, as suggested.

REVIEWER #4

This paper describes a very interesting and important topic of how individual variation in connectome organization relates to pathophysiology in a group of developing adolescents and adults. The paper is very well written.

We appreciate the reviewer’s summary and positive appraisal of the study. As detailed below, we are very happy to integrate the helpful feedback offered.

1. Perhaps the introduction could be a little bit shortened.

We have streamlined the introduction as suggested.

2. How was the discovery and replication dataset formed? With the two sets originating from the same dataset, statistically splitting the 2 sets is not a big advantage (or difference).

We agree with the reviewer that having an independently acquired dataset for validation would be the gold standard. However, the analytic approach used here requires equivalent clinical data across samples for comparability. At present, this is a substantial obstacle, as no other studies have used the GOASSESS interview besides the PNC. This underscores important next steps, which we highlight in the revised Limitations section:

It should be noted that our replication sample was constructed from the PNC data, rather than using an independently acquired dataset. This approach was dictated by the lack of correspondence with clinical instruments used in other large-scale developmental imaging studies. This limitation underscores the need for harmonization of not just imaging data but also clinical measures across studies moving forward.

Given this limitation, we used a split-sample approach for discovery and replication in the PNC. Detailed information regarding sample construction was provided in the Online Methods and in **Supplementary Figure 1**. These are included below for reference:

To create two independent samples for discovery and replication analyses, we performed a random split of the remaining 1600 participants using the CARET package in R. Specifically, using the function `CreateDataPartition`, a discovery sample

(n=1069) and a replication sample (n=531) were created that were stratified by overall psychopathology (**Supplementary Fig. 1**). The two samples were confirmed to also have similar distributions in regards to age, sex, and race (**Fig. 1b**), as well as motion (**Supplementary Fig. 2**).

Supplementary Figure 1 | Sample Construction The cross-sectional sample of the Philadelphia Neurodevelopmental Cohort (PNC) has 1601 participants in total. Excluding the one missing clinical data, 1600 participants were randomly stratified into a discovery (n=1069) and a replication (n=531) sample. Applying health, structural and functional imaging quality exclusion criteria, 663 and 336 subjects were included in the final discovery and replication samples, respectively.

3. An in-text description of the 111 clinical items (or a categorization of this) would be very nice. Now ‘clinical items’ remain a little bit vague.

We agree, and have revised the results section to reflect this suggestion on Page 6:

The input data thus consisted of 111 clinical items and 3,410 unique functional connections (**Figure 2 d,e**). The clinical items were drawn from the structured GOASSESS interview (Calkins et al., J. Child Psychology & Psychiatry, 2015), and covers a diverse range of psychopathological domains, including mood and anxiety disorders, psychosis-spectrum symptoms, ADHD, and other disorders (see details in **Supplementary Table 2**).

4. Page 6, last paragraph: ‘Of these seven canonical variates, three were significant (Pearson correlation $r = 0.71$, PF DR < 0.001; $r = 0.70$, PF DR < 0.001, $r = 0.68$, PF DR < 0.01, 18 respectively) (Fig. 2b), with the fourth showing a trend towards significance ($r = 0.68$, PF DR = 0.07, 19 Puncorrected = 0.04).’ this paragraph was a little bit unclear to me. What was examined here with the 7 variates, and what was ‘significant’? What was tested against what and how?

We are happy to clarify. As detailed in the Online Methods, sCCA produces as many modes of variation as the size of the smallest input data matrix. Thus, in this case, sCCA yielded 111 modes of covariation, based on the dimensions of the clinical data. Which we examined these modes of covariation using a scree plot (**Figure 2A**). Of these 111 modes, we selected seven modes for further analysis based on the elbow of the variance explained in the scree plot, a common criterion for filtering this data. These seven modes were evaluated for significance using permutation testing, which was devised to test whether the correlation between the clinical features and the imaging features was greater than would be expected using permuted data. Specifically, the permutation test was performed by holding the connectivity matrix constant, while shuffling the rows of the clinical features, essentially assigning one subject's clinical symptoms to another subject's functional connectivity features, thus creating a null distribution of canonical correlations using these permuted input data (see **Supplementary Figure 4**). The level of significance of each of the seven canonical variates was then assessed by comparing the actual correlation against the null distribution and then was corrected for multiple comparisons using the False Discovery Rate ($Q < 0.05$).

5. Overall, I found the methodology and statistical analysis a little bit hard to follow. I appreciate and understand the complexity of this approach, but explaining it in a 'simple' matter would be very interesting, and make it more clear to the reader which is less advanced in statistical procedures.

We value this feedback, and have made an effort to comprehensively revise the methods and results section in order to make the approach more accessible for a wide range of readers.

6. The authors describe that the networks were restricted to the top 10 percent of most variable connections. I have a major concern with this, as the most variable connections are potentially also the most 'noisy' connections as they are the least consistent across subjects. Noisy connections can have a strong impact on graph metrics (see the recent work of Zaleksy 2016 Neuroimage and Heuvel 2016 Neuroimage). I think it is vital to show that similar results are also found when no 'thresholding' of some sort is performed. Can the authors include such an analysis as validation?

We agree, and accordingly did examine approaches to thresholding as part of the supplementary analyses included in the original manuscript (see **Supplementary Figure 6A**). It should be noted that sCCA performance would be substantially degraded without any dimensionality reduction or thresholding, as the regularization would become unstable with such a large number of features (~35,000) for the (relatively small) number of samples ($n=663$). However, an alternative approach that does not require selection of a specific threshold is to first reduce the dimensionality of

the imaging data using principal components analysis (PCA) prior to testing with CCA. PCA has been widely used in conjunction with CCA in the past (for example, see Smith et al., *Nature Neuroscience* 2015). As shown in **Supplementary Figure 6C**, repeating the analysis using PCA and un-thresholded matrices produces highly convergent results, including dimensions corresponding to mood, psychosis, fear, and externalizing behavior.

7. The examination was limited to functional connectivity, and extension of findings to structural connectivity (of which is available in this dataset) would be very interesting.

While we agree that it would be very interesting to incorporate structural connectivity, it is beyond of scope of the current paper. However, we note this fertile direction for future research in the revised Discussion section:

Finally, our current analysis only considered functional connectivity and clinical psychopathology. Future research could incorporate rich multi-modal imaging data, cognitive measures, and genomics.

8. Parcellation: one specific parcellation atlas was chosen, but common practice is to show robustness of findings when other parcellation atlases are used for network reconstruction. Can the authors briefly show findings across other atlases /other resolutions?

We agree, and in the original manuscript included analyses using the parcellation provided by Gordon et al. (*Cerebral Cortex*, 2016) as an alternative. As detailed in **Supplementary Figure 6B**, this analysis yielded highly convergent results.

9. The term 'dysconnectivity' was used, but given the notion that the included subjects did not include actual 'patients', perhaps variation in connectivity is a better term? Also in Figure 4 the authors use 'increased' and 'decreased' but perhaps, 'lower'/'higher' is a better term to use in this context?

As suggested, we have now changed “dysconnectivity” to “connectivity” where appropriate. Furthermore, we have also updated the text in Figure 4 to “lower” and “higher” to replace “decreased” and “increased”.

10. Figure 2. The plots are very nice, but perhaps a little bit misperceiving. The sCCA is designed to find this type of correlations, so the results presented are the direct consequence of the sCCA analysis. Perhaps a good null-condition in this context is to shuffle the FC weights across the selected edges in the network, and then re-perform the sCCA analysis. How many significant axis were found in this null-condition, and are

the number of sCCA patterns found now significantly exceeding what one would find in the null-condition?

We are happy to clarify. As detailed in our response to point 4 above, the null distribution illustrated in **Figure 3** is obtained by holding the connectivity matrix constant, while shuffling the rows of the clinical features, essentially assigning one subject's clinical symptoms to another subject's functional connectivity features, thus creating a null distribution of canonical correlations using these permuted input data (see **Supplementary Figure 5**). Such permutation procedures have been widely used in prior work (Smith et al., *Nature Neuroscience* 2015; Misc et al, *Cerebral Cortex*, 2016). An important advantage of permuting the participants is that it allows one to keep the covariance structures within both clinical and imaging data.

Reviewers' comments:

Reviewer #1 (Remarks to the Author):

Here is my updated take on this paper.

I reaffirm my initial assessment that the study:

- a) is fundamentally a high-quality and high-impact work.
- b) stands out positively from work of similar type and scope.
- c) will fit right in at Nature Communications.

Having said this, I note that the authors have not adequately addressed the concerns I previously raised on the effects of their aggressive preprocessing strategy. Let me elaborate why it is important to consider these concerns in more detail. Fundamentally, the BOLD signal is incompletely understood, and the field's assumptions of what represents signal and noise are somewhat arbitrary and ad hoc. This makes it difficult to arrive at any sort of consensus (e.g. Murphy and Fox in their "towards consensus" paper have essentially agreed to disagree), and in turn to evaluate the authors' arguments for not doing additional verifications. In fact, the authors have noted as much in Ciric et al. (2017),

"Our primary benchmark of confound regression performance assumes that mitigation of the relationship between QC (participant motion) and FC (i.e., the imaging measurement) is desirable. To the degree that in-scanner motion itself represents a biologically informative phenotype, this approach will mistake signal for noise. Indeed, prior data suggests that this may sometimes be the case. For example, Zeng et al. (2014) found specific changes in connectivity for participants who had generally high levels of motion, even on scans where motion was low."

I understand that the present group has done much pioneering work in motion correction. I have read these papers and appreciate the contributions. But motion is not the be-all and end-all confound in fMRI, and in psychiatric populations specifically, vigilance levels of each subject may represent a more fundamental source of variance that drives the observed effects. Of course, in some cases high vigilance will drive high motion so these confounds are not independent.

One reason for the popularity of global signal regression is its simple one-step strategy for eliminating all sources of potential confound from the data (physiological, motion, vigilance, etc). Yet, global signal regression is fundamentally a blunt tool that deforms the signal in ways that differ from subject to subject and are impossible to predict. Liu et al. (2017) give a good overview for the sources of "signal" and "noise" that are removed with this procedure:

<https://www.ncbi.nlm.nih.gov/pubmed/28213118>

In this context, here are my specific recommendations for considering the effects of preprocessing:

- 1) Find the "least noisy" ($N \sim 200$) and "most noisy" ($N \sim 200$) one thirds of the $N=663$ training dataset.
- 2) Compare the differences in effects on these subgroups processed with a less aggressive 16-parameter or 24-parameter strategy that does not include global signal regression.

These analyses should not be done to refute the main results, but rather to provide a greater understanding of the extent to which the effects are in fact driven by more basic sources of the signal, such as vigilance or physiology. As already mentioned, such sources are important in their own right, are not necessarily artifacts, and provide clearer neurobiological underpinnings of higher-order explanations.

Reviewer #2 (Remarks to the Author):

The authors have done a commendable job in addressing reviewer concerns and revising the manuscript. I have no further comments.

Reviewer #3 (Remarks to the Author):

I was already impressed by the manuscript during the first round and am reassured by this revised version and the new analyses provided. All my comments have been satisfactorily addressed. The authors should be complimented for their hard work.

Please just ensure that added texts and references are identical between the rebuttal and the main text (e.g. p16 of the manuscript).

I have no further comment.

Reviewer #4 (Remarks to the Author):

The authors have nicely addressed the comments raised.

As final remarks: Splitting a large sample in 2 and then use one as a replication set does not necessarily bring statistical validation. It is a common misassumption that this is a good thing to do (and it may sound more robust than just using 1 sample). However, as far as I know, having 2 samples is just providing the same effect but twice with a higher p-value. Grouping the total set (which was already 1 set to begin with) will just give you lower p-values on 'true' effects. A similar results against 'false positives' could thus also be derived by using a more strict alpha level. It might be worth to mention something about in the text. A comment on this as a limitation (as compared to using 2 independent sets) might be in place.

In response to my point 10, the authors nicely mentioned that clinical scores were shuffled. This is an alternative approach to using a correlation value (when data is normally distributed the effects will be the same as doing a regression). However, shuffling of the matrix (which the authors here do not do), might be a more strict method, as it now keeps global and binary structure the same, BUT makes the metric of interest (the distribution of FC) random in the null conditions. Shuffling the weights thus keeps some properties of none interest (overall FC etc) in the null model, effectively including properties of non-interest as covariates in your analysis (which is not done when just shuffling the clinical values). As I mentioned in my previous response, I would thus be interested to also see the results of this type of analysis, as it is the more common way of analysing effects in brain network studies. It would be great if the authors have time/space to add this analysis for final verification.

REVIEWER #1

Here is my updated take on this paper.

I reaffirm my initial assessment that the study:

- a) is fundamentally a high-quality and high-impact work.*
- b) stands out positively from work of similar type and scope.*
- c) will fit right in at Nature Communications.*

We thank Reviewer #1 for the positive appraisal of our work, and are happy to integrate the thoughtful feedback offered.

Having said this, I note that the authors have not adequately addressed the concerns I previously raised on the effects of their aggressive preprocessing strategy. Let me elaborate why it is important to consider these concerns in more detail. Fundamentally, the BOLD signal is incompletely understood, and the field's assumptions of what represents signal and noise are somewhat arbitrary and ad hoc. This makes it difficult to arrive at any sort of consensus (e.g. Murphy and Fox in their "towards consensus" paper have essentially agreed to disagree), and in turn to evaluate the authors' arguments for not doing additional verifications. In fact, the authors have noted as much in Ciric et al. (2017),

"Our primary benchmark of confound regression performance assumes that mitigation of the relationship between QC (participant motion) and FC (i.e., the imaging measurement) is desirable. To the degree that in-scanner motion itself represents a biologically informative phenotype, this approach will mistake signal for noise. Indeed, prior data suggests that this may sometimes be the case. For example, Zeng et al. (2014) found specific changes in connectivity for participants who had generally high levels of motion, even on scans where motion was low."

I understand that the present group has done much pioneering work in motion correction. I have read these papers and appreciate the contributions. But motion is not the be-all and end-all confound in fMRI, and in psychiatric populations specifically, vigilance levels of each subject may represent a more fundamental source of variance that drives the observed effects. Of course, in some cases high vigilance will drive high motion so these confounds are not independent.

One reason for the popularity of global signal regression is its simple one-step strategy for eliminating all sources of potential confound from the data (physiological, motion, vigilance, etc). Yet, global signal regression is fundamentally a blunt tool that deforms the signal in ways that differ from subject to subject and are impossible to predict. Liu et al. (2017) give a good overview for the sources of "signal" and "noise" that are removed with this procedure:<https://www.ncbi.nlm.nih.gov/pubmed/28213118>

In this context, here are my specific recommendations for considering the effects of preprocessing:

- 1) Find the "least noisy" (N~200) and "most noisy" (N~200) one thirds of the N=663 training dataset.*
- 2) Compare the differences in effects on these subgroups processed with a less aggressive 16-parameter or 24-parameter strategy that does not include global signal regression.*

These analyses should not be done to refute the main results, but rather to provide a greater understanding of the extent to which the effects are in fact driven by more basic sources of the

signal, such as vigilance or physiology. As already mentioned, such sources are important in their own right, are not necessarily artifacts, and provide clearer neurobiological underpinnings of higher-order explanations.

Following the reviewer's suggestions, we reprocessed the resting-state fMRI data without regressing out the global signal. Instead, we used 12 parameter + *aCompCor*, which is one of the best performing reprocessing procedures to correct for motion without GSR (see Behzadi et al., 2007; Muschelli et al., 2014; Ciric et al., 2017; Parkes et al., 2018). Subsequently, we constructed network matrices of functional connectivity using the reprocessed imaging data in the discovery cohort (n=663). We followed the same procedures as in the main analysis and selected the first five canonical variates for further analysis based on covariance explained (**Figure 1A**). All five canonical correlations were statistically significant by permutation testing with FDR correction ($q < 0.05$), of which the first four re-capitulate the dimensions reported in the main analysis (**Figure 1B**). Furthermore, a fifth dimension – corresponding to OCD-spectrum symptoms – was significant as well.

Figure 1 | Linked dimensions of psychopathology in re-processed data without global signal regression. (a) The first five canonical variates were selected for further analysis based on covariance explained. Dashed line marks the average covariance explained. **(b)** The original four dimensions – psychosis, mood, fear, and externalizing behavior – and a fifth dimension (corresponding to OCD-spectrum symptoms) were significant by permutation testing. Corresponding variates are circled in panel (a). Error bars denote standard error. *** $P_{FDR} < 0.001$, ** $P_{FDR} < 0.01$, * $P_{FDR} < 0.05$.

To specifically compare the differences in effects on those with the most motion and those with the least motion (n=199, and n=200 respectively, **Figure 2a**), we calculated the correlation coefficient between the connectivity score processed with GSR and the connectivity score processed without GSR in each motion group. Similarly, we also calculated the correlation coefficient between the clinical score processed with and without GSR in each motion group. All correlations were statistically significant (all $P < 2.2 \times 10^{-16}$) with correlation coefficients ranging from 0.38 to 0.68 for connectivity scores, and from 0.77 to 0.99 for clinical scores (**Figure 2b**). This result demonstrates that our findings are robust to the methodological choice of performing global signal regression.

Figure 2 | Comparison of effects of low and high motion subjects processed with and without global signal regression. (a) Histogram of subject in-scanner motion in the discovery cohort (n=663), of which those with the lowest motion (<0.041mm, n = 200) and those with the highest motion (>0.077mm, n = 199) were selected for the comparison of their CCA dimensional scores processed with and without

GSR. **(b)** We calculated the correlation coefficient between the CCA dimensional scores (i.e. connectivity and clinical scores) processed without GSR (x axis) and those processed with GSR (y axis) in each motion group for each of the four canonical dimensions. All correlation coefficients were highly significant ($P < 2.2 \times 10^{-16}$).

REVIEWER #2:

The authors have done a commendable job in addressing reviewer concerns and revising the manuscript. I have no further comments.

We thank Reviewer #2 for their time and helpful comments.

REVIEWER #3:

I was already impressed by the manuscript during the first round and am reinsured by this revised version and the new analyses provided. All my comments have been satisfactorily addressed. The authors should be complimented for their hard work.

Please just ensure that added texts and references are identical between the rebuttal and the main text (e.g. p16 of the manuscript).

I have no further comment.

We thank Reviewer #3 for their useful feedback and positive appraisal of our work. In the revised version, we now ensure that all references are consistently cited as in the response.

REVIEWER #4:

The authors have nicely addressed the comments raised.

We thank Reviewer #4 for the positive comments, and for his/her earlier suggestions, which have significantly improved the work.

As final remarks: Splitting a large sample in 2 and then use one as a replication set does not necessarily bring statistical validation. It is a common misassumption that this is a good thing to do (and it may sound more robust than just using 1 sample). However, as far as I know, having 2 samples is just providing the same effect but than twice with a higher p-value. Grouping the total set (which was already 1 set to begin with) will just give you lower p-values on 'true' effects. A similar result against 'false positives' could thus also be derived by using a more strict alpha level. It might be worth to mention something about in the text. A comment on this as a limitation (as compared to using 2 independent sets) might be in place.

We appreciate the feedback and have now added the limitation in the revised discussion:

Discussion, Page 19:

Fourth, our replication sample was constructed from the PNC data. Using an independently acquired dataset to validate our findings would provide evidence of greater generalizability than splitting the original data into two samples. However, this approach was dictated by the lack of correspondence with clinical instruments used in other large-scale developmental imaging studies. This limitation underscores the need for harmonization of not just imaging data but also clinical measures across studies.

In response to my point 10, the authors nicely mentioned that clinical scores were shuffled. This is an alternative approach to using a correlation value (when data is normally distributed the effects will be the same as doing a regression). However, shuffling of the matrix (which the authors here do not do), might be a more strict method, as it now keeps global and binary structure the same, BUT makes the metric of interest (the distribution of FC) random in the null conditions. Shuffling the weights thus keeps some properties of none interest (overall FC etc) in the null model, effectively including properties of non-interest as covariates in your analysis (which is not done when just shuffling the clinical values). As I mentioned in my previous response, I would thus be interested to also see the results of this type of analysis, as it is the more common way of analysing effects in brain network studies. It would be great if the authors have time/space to add this analysis for final verification.

We appreciate the feedback. There are two distinct ways that we could shuffle the connectivity feature matrix, either by randomizing the connectivity adjacency matrix, hence building a null network model or by permuting the rows of connectivity feature matrix, hence disrupting the subject correspondence between the connectivity and clinical features.

It is important to note that the first approach, shuffling the original adjacency matrix while preserving the edge weight distribution, would disrupt the covariance structure. Because CCA relies on the covariance structure of the input data to capture the sources of variation that are common to both datasets, it is sensitive to the true rank of the input data. To illustrate this effect, we generated simulated data by synthesizing two matrices with sizes that are identical to that of the original data ($X_{663 \times 3410}$ and $Y_{663 \times 111}$) (simulation CCA data synthesis as implemented in Gao et al., *The Annals of Statistics*, 2017). Then we reduced the rank of the X matrix by duplicating random columns while maintaining the apparent dimension of 663x3410. We calculated the effective rank of the X matrix, which is defined as the ratio of the sum of all singular values divided by the largest singular value (Roy and Vetterli, *IEEE Singal Processing Conference*, 2007). As the effective rank of the X matrix increases, the estimated canonical correlation of the first component also increases (**Figure 3**). This effect demonstrates that while the apparent dimensionality of the matrices remains the same, the canonical correlation depends on the effective rank of the matrix.

Figure 3 | CCA is sensitive to the effective rank of the input matrix. We generated simulated data with the same apparent dimensions as the original data but with varying effective rank. As the effective rank increases, the canonical correlation also increases.

This is relevant because shuffling the connectivity matrix by randomizing the original adjacency matrix artificially increases the effective rank. For example, the original connectivity matrix has an effective rank of 145, whereas the randomized network has an effective rank of 432 (insert in **Figure 4**). Alternatively, one can also appreciate this effect by examining the scree plot of the principal components of each matrix (**Figure 4**), where the original data requires fewer principal components to explain the same amount of variance than the shuffled data. Such an artificial increase in the effective rank of the input matrix by randomizing by columns also inflates the estimated canonical correlation of the null distribution (as shown in **Figure 3**). Intuitively, this process creates a mis-match between the rank of the real data and the rank of the null model, making the null model inappropriate for hypothesis testing. It is also worth noting that this null model has not been typically used in the CCA literature for this reason.

Figure 4 | Shuffled adjacency matrix inflates the rank of the input data. We conducted principal component analysis on both the original connectivity matrix (blue) and the connectivity matrix shuffled by randomizing the adjacency matrix (red). Compared to the original data, the shuffled matrix would require

more components to explain the same amount of the matrix variance. Specifically, we calculated the effective rank of each matrix (**insert**) and found that the shuffled matrix has artificially inflated rank compared to the original data. The increased rank in the shuffled matrix has the consequence of inflating the estimated canonical correlation of the null model (see **Figure 3**).

Alternatively, we conducted the permutation procedure by shuffling the connectivity feature matrix *by rows* (i.e., subjects), instead of randomizing the adjacency matrix as shown above. Critically, permuting the connectivity feature matrix by row would disrupt the linkage between subjects' connectivity and clinical features without influencing the effective rank of the input data. Using these permuted matrices, we created null distributions for each dimension (**Figure 5**). Consistent with the findings reported in the main manuscript, three psychopathological dimensions (mood, psychosis, and fear) were identified and found to be significant in permutation testing, with a fourth dimension displaying a trend toward significance after FDR correction for multiple comparisons. This analysis demonstrated that our findings of linked dimensions of connectivity and psychopathology were robust to shuffling either the clinical matrix (as done in the main manuscript) or the connectivity matrix.

Figure 5 | Shuffling the connectivity matrix by subjects produces results consistent with the original findings. We created a null distribution of the canonical correlations by shuffling the connectivity matrix by subjects, thereby disrupting the link between connectivity and clinical data. Consistent with the main analysis, three linked dimensions (mood, psychosis, and fear) were statistically significant after FDR correction, with a fourth dimension (externalizing behavior) displaying a trend towards significance.

REVIEWERS' COMMENTS:

Reviewer #1 (Remarks to the Author):

As I suggested, the authors have performed additional analyses to look at the effects of global signal regression on their results. However, it seems that these analyses have not actually been included in the body or SI of the manuscript, and so seemed to have been performed solely to satisfy this specific reviewer, rather than to inform the broader readership of Nature Communications.

It is important to note here that global signal regression remains an important and polarizing methodological question, which has bothered the field for some time; i.e. it is more than just a pet peeve of a reviewer.

In this context, I am confused why the authors did not include these additional analyses in the body or manuscript of the SI.

Reviewer #4 (Remarks to the Author):

The authors have (very nicely) addressed my comments. I like to congratulate them with this very interesting study!

REVIEWER #1

As I suggested, the authors have performed additional analyses to look at the effects of global signal regression on their results. However, it seems that these analyses have not actually been included in the body or SI of the manuscript, and so seemed to have been performed solely to satisfy this specific reviewer, rather than to inform the broader readership of Nature Communications.

It is important to note here that global signal regression remains an important and polarizing methodological question, which has bothered the field for some time; i.e. it is more than just a pet peeve of a reviewer.

In this context, I am confused why the authors did not include these additional analyses in the body or manuscript of the SI.

We have now incorporated the additional analysis of GSR effects in the Supplementary Information (**Supplementary Figure 3** and **Supplementary Figure 4**). We thank Reviewer #1 for their time and helpful comments.

REVIEWER #4

The authors have (very nicely) addressed my comments. I like to congratulate them with this very interesting study!

We thank Reviewer #2 for their time and helpful comments.